



# Benchmarking convection-permitting climate simulations for hydrological applications: A comparative study of WRF-SAAG and observation-based products

Sofía Segovia[1], Pablo A. Mendoza[1,2], Miguel Lagos-Zúñiga[3], Lucía Scaff[4], and Andreas Prein[5]

[1]Department of Civil Engineering, Universidad de Chile, RM, Santiago, Chile
[2]Advanced Mining Technology Center (AMTC), Universidad de Chile, RM, Santiago, Chile
[3]Departamento de Obras Civiles, Universidad Técnica Federico Santa María, RM, Santiago, Chile
[4]Department of Geophysics, Universidad de Concepción, Bio-Bio, Concepción, Chile
[5]Institute for Atmospheric and Climate Science, ETH Zürich, 8092 Zurich, Switzerland

**Correspondence:** Pablo A. Mendoza (pamendoz@uchile.cl)

**Abstract.** Over the last years, significant progress has been made in the development of convection-permitting climate models (CPCMs), especially for improving precipitation modeling in regions with complex terrain. Recently, the South American Affinity Group (SAAG) developed a novel high-resolution dataset — hereafter referred to as the WRF-SAAG dataset — by dy-
namically downscaling the ERA5 reanalysis using the Weather Research and Forecasting (WRF) model over South America for the period 2000–2021. In this paper, we evaluate the quality of WRF-SAAG daily precipitation and temperature simulations using observations from meteorological stations over continental Chile for the period 2001–2018, and present comparisons against two gridded meteorological products – CR2MET and RF-MEP – which are based on in-situ meteorological station measurements and have been widely used for hydrometeorological applications in this region. We found that, although the
precipitation products correctly replicated the percentage correct ($PC$) of observed events and non-events ($PC \geq 0.64$), detection accuracy varied within each Chilean macrozone –defined by latitudinal bands – with worse performance in the Far North (between 17.5 – 26°S) and Patagonia (between 43.7 – 56°S) — median Critical Success Index ($CSI$) < 0.49 for events > 5 mm/d— compared to the central region ($CSI \geq 0.44$ for events > 5 mm/d). The evaluation of daily precipitation and extreme temperatures against station observations using Tang's Kling-Gupta efficiency ($KGE_T$) and its components reveals that all
datasets performed better in reproducing precipitation in rainy regions (median $KGE_T \geq 0.65$ in the Southern macrozone), while in arid areas such as the Near North during summer, the median $KGE_T$ was negative. The CR2MET product consistently provided the best performance metrics for extreme precipitation and temperature, partly because it includes information from the stations used for evaluation. Finally, the application of the TUW hydrological model shows that WRF-SAAG simulations achieved runoff estimations comparable to the best observation-based products, with the best metrics obtained in the Southern
macrozone, where the median objective function ($OF$) —defined as the average of $KGE'$ and $KGE'(1/q)$ — remains above 0.87 (0.67) during the calibration (evaluation) period. More broadly, the results presented here show that – despite some remaining challenges in arid climate regions – kilometer-scale climate models can deliver information of a quality comparable to that of observation-based products for hydrological applications in Chile.



## 1 Introduction

Spatially distributed meteorological datasets are not only essential to understand weather and climate patterns, but also to characterize hydrological systems in various contexts (e.g., water balances, extreme events) and make predictions for water resources management and planning. Hence, the evaluation (e.g., Rasmussen et al., 2011; Mendoza et al., 2015) and inter-comparison (e.g., Zambrano-Bigiarini et al., 2017; Henn et al., 2018; Newman et al., 2019) of such datasets is critical for understanding their potential for different applications, especially in areas with complex topography where the scarcity of

observations (e.g., Viviroli et al., 2011; Muñoz et al., 2024), the lack of long-term records (e.g., Barrios et al., 2018; Serrano-Notivoli and Tejedor, 2021), and precipitation undercatch (e.g., Rasmussen et al., 2012; Prein and Gobiet, 2017) introduce large uncertainties in the estimation of hydrometeorological variables.

In recent decades, several gridded meteorological datasets have been produced worldwide for catchment-scale, regional, and global-scale applications. Although spatial interpolation schemes based on in-situ measurements and topographic de-

scriptors have been the standard approach (Daly et al., 1994, 2008; Clark and Slater, 2006; Isotta et al., 2014; Newman et al., 2015, 2020), their accuracy heavily depends on the spatial density of stations, which is typically higher in valleys and populated areas. Hence, satellite-based products have become an attractive alternative in sparse data regions, especially for rainfall (e.g., Kidd and Huffman, 2011; Lockhoff et al., 2014; Funk et al., 2015), though they tend to underestimate precipitation intensities, overestimate their frequency (Scheel et al., 2011; Katiraie-Boroujerdy et al., 2013), and depend on the retrieval algorithm (Bart-

sotas et al., 2018). Further, satellite-based precipitation estimates have particularly large uncertainties and biases in mountain regions (Derin and Yilmaz, 2014). Dynamical climate simulations have also been useful in producing global-coverage historical time series at moderate ($\sim$0.25-0.75°) resolutions, contributing to an improved understanding of mesoscale meteorology (e.g., Torma et al., 2015; Rummukainen, 2016; Vautard et al., 2021). In particular, atmospheric reanalysis products are generated with numerical weather prediction (NWP) models that assimilate a variety of observations, resulting in historical time

series that contain a full set of meteorological variables (e.g., Saha et al., 2010; Rienecker et al., 2011; Hersbach et al., 2020), which can be used for detailed hydrological modeling analyses. However, these products still feature large biases, especially in mountainous regions, because of the inherent simplification of mesoscale processes in a smoothed topography. Finally, several gridded meteorological products have been developed by blending different data sources of meteorological variables, including in situ measurements, radar data, satellite products, reanalysis products and NWP model output. For example, Verdin et al.

(2015) developed a product by combining satellite estimates with rain gauge observations using a Bayesian kriging approach; Beck et al. (2017) produced the global Multi-Source Weighted-Ensemble Precipitation (MSWEP) dataset by merging gauge, satellite, and reanalysis data; and Yin et al. (2021) proposed a three-stage blending approach integrating multiple satellite and reanalysis products with gauge data.

Numerous studies have assessed the suitability of gridded precipitation and temperature datasets to provide reliable hydro-

logical model simulations (e.g., Kouakou et al., 2023; Evin et al., 2024; Gebrechorkos et al., 2024; Jahanshahi et al., 2024). Overall, previous work suggests that observation-based meteorological products generally outperform other datasets in runoff estimation, and that bias correction using observational data improves the accuracy of runoff simulations (e.g., Kouakou et al.,



2023; Jahanshahi et al., 2024). However, the reliability of observation-based products is inherently constrained by the spatial density of meteorological stations (Terink et al., 2018; Herrera et al., 2019). Lundquist et al. (2019) concluded that, in
mountainous regions, the low density of meteorological stations relative to the spatial variability of precipitation —particularly snowfall —, combined with the systematic undercatch of rain gauges, limits the representativeness of gridded observational products. As a result, high-resolution atmospheric models can outperform gridded observational products in capturing total precipitation over complex terrain. Additionally, these models offer a physically consistent and spatially continuous representation of precipitation, making them a viable alternative for hydrological modeling applications.

Because of its complex topography and large hydroclimatic diversity (Sarricolea et al., 2017; Aceituno et al., 2021) continental Chile (∼17° - 57°S) is an interesting domain for the development and evaluation of gridded meteorological products for hydrometeorological applications. For example, Boisier et al. (2018) created the gridded meteorological product CR2MET based on the combination of in-situ observations and ERA5 (Hersbach et al., 2020) reanalysis outputs, whereas Baez-Villanueva et al. (2020) developed a merging procedure for precipitation estimation, which consists of the combination of observational data,
meteorological products (e.g., ERA5 reanalysis) and topographic covariates. While these observation-based datasets have been widely used for different applications (e.g., Hernandez et al., 2022; Murillo et al., 2022 in the case of CR2MET, and Chen et al., 2022; Al-Saeedi et al., 2024 in the case of RF-MEP) there is limited information about their performance. The far north (above 26°S) and far south (below 43.7°S) of Chile are regions of special interest, since the station density is considerably lower and, therefore, reanalysis products become critical for the development of gridded meteorological products and, in particular,
precipitation estimates. Further, there is a high disagreement among CR2MET, RF-MEP, and ERA5 precipitation estimates over those areas. In the far north (above 26°S), annual differences reach up to 117 mm between products (CR2MET: 63 mm, RF-MEP: 40 mm, ERA5: 157 mm), while in the far south (below 43.7°S), discrepancies are even larger, reaching up to 2203 mm (CR2MET: 1888 mm, RF-MEP: 815 mm, ERA5: 3018 mm) (Baez-Villanueva et al., 2021).

In the Andes, the performance of meteorological products varies considerably, with substantial differences between precipitation estimates north of 24°S and south of 35°S, and among temperature estimates between 27°S and 35°S (Schumacher et al.,
2020b). Zambrano-Bigiarini et al. (2017) assessed seven satellite-based rainfall products, finding that most of them perform better between 32°S and 43°S at elevations below 1000 m a.s.l., while the poorest performance was obtained for elevations above 2000 m a.s.l. On the other hand, Zambrano et al. (2017) showed that satellite products generally capture the rainiest months more accurately than dry months, especially during the Austral winter (JJA), except north of 28°S, where rainfall is
concentrated in summer (DJF).

During the last decade, convection-permitting climate models (CPCMs) have become increasingly popular (Lucas-Picher et al., 2021) because they offer an enhanced representation of precipitation (e.g., Fosser et al., 2020), and do not rely on cumulus parametrizations – detected as an important source of errors in regional climate modeling –, improving land-atmosphere interactions (Prein et al., 2015). CPCMs also offer the opportunity to advance hydrometeorological understanding at kilometer-
scale resolution, and have been used for a myriad of purposes, including snowpack analysis (Ikeda et al., 2021), cloud band detection (Zilli et al., 2024), and flood studies (Li et al., 2022) over continental domains (e.g., Liu et al., 2025).



Recently, the South American Affinity Group (SAAG) – supported by the National Science Foundation (NSF) National Center for Atmospheric Research (Dominguez et al., 2024) – generated an unprecedented high-resolution gridded meteorological dataset for South America using the Weather Research and Forecasting (WRF; Skamarock et al., 2019) model. This dataset – hereafter referred to as WRF-SAAG – offers new research opportunities for the region, characterized by scarce and discontinuous observational records (Condom et al., 2020). The WRF-SAAG precipitation simulations revealed high spatial correlations with observed annual averages (0.65 -0.8), and 0.95 correlation coefficient for mean annual 2-m temperature compared with observational and reanalysis products over the entire South American continent (Dominguez et al., 2024). Hourly evaluations indicate that the WRF-SAAG simulations better reproduce the diurnal cycle and heavy hourly precipitation over the East of Brazil (Dominguez et al., 2024). However, some regions feature annual biases up to 400 mm/year (Liu et al., 2025). Despite other studies having evaluated the representation of cloud bands (Zilli et al., 2024) and tracking mesoscale systems (Núñez Ocasio and Moon, 2024; Prein et al., 2024; Rehbein et al., 2025), to the best of our knowledge, a hydrologically-oriented long-term evaluation has not been conducted, except for an individual watershed in the arid western Andes ( ∼28°S, Sanhueza, 2024).

In this paper, we present a regional assessment of WRF-SAAG precipitation and temperature simulations across continental Chile. We first evaluated the simulation's accuracy at the station scale, and then explored its potential for broader applications. To this end, we used in situ observations from meteorological stations as the reference dataset. Additionally, we compared the performance of WRF-SAAG precipitation simulations against the CR2MET (Boisier et al., 2018) and RF-MEP (Baez-Villanueva et al., 2020) gridded meteorological products, while temperature simulations were benchmarked against CR2MET estimates. Finally, we assess the suitability of the analyzed datasets for hydrological modeling across a suite of 44 near-natural catchments with varying hydroclimatic regimes. We stress that the WRF-SAAG simulation was not designed to replicate individual weather events, but rather to represent hydroclimatic features over South America, aiming to improve physical understanding and support decision-making in a changing climate. Additionally, CR2MET and RF-MEP are not fully independent from the station observations used as reference in this evaluation, since both products incorporate in situ data in their construction. This study provides valuable insights into the strengths, limitations, and applicability of CPCM outputs for hydrological studies in regions with complex topography and limited ground-based observations.

## 2 Study domain

The study area is continental Chile (Fig. 1), which spans more than 4200 km from north (17°29'S) to south (55°58'S) and is bounded by the Pacific Ocean in the west and the Andes Cordillera in the East. This domain encompasses four main geographical units from West to East: the Coastal Plains, the Coastal Range, the Intermediate Depression, and the Andes Cordillera. Further, the Chilean Water Directorate (DGA by its acronym in Spanish) defines five major macroclimatic zones for water resources management and planning: Far North (17.5 – 26°S); Near North (26 – 32.18°S); Central Chile (32.18 – 36.4°S); Southern Chile (36.40 – 43.7°S); and Austral Chile (43.7 – 56°S). These macrozones span a wide variety of climates, with very specific features across the austral seasons: summer (DJF), autumn (MAM), winter (JJA), and spring (SON).



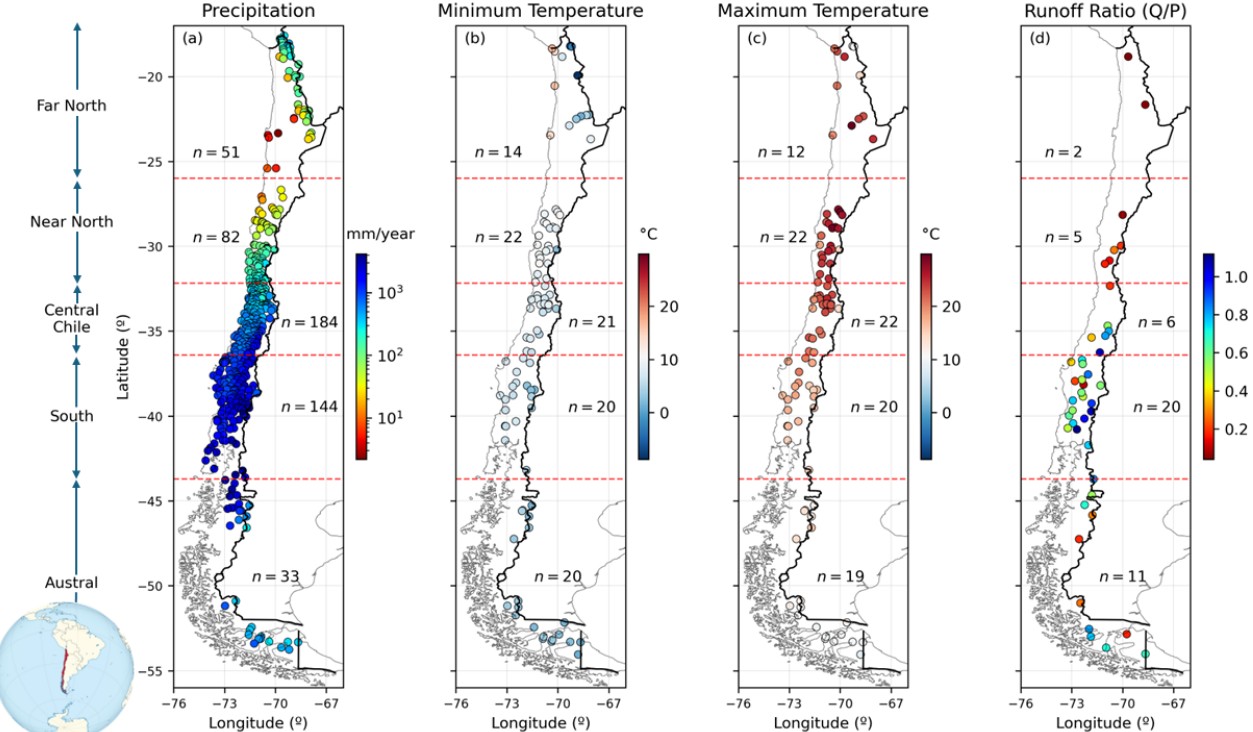

**Figure 1.** Climatological averages based on in-situ measurements for (a) annual precipitation, (b) minimum daily temperature and (c) maximum daily temperature. (d) Mean annual runoff ratio of the selected basins, calculated with the observed runoff and the precipitation in the basin obtained from CR2MET (v2.5), along with the delineation of macrozones and the location of selected stations. All hydroclimatic indices are computed for the period April/2001-March/2018, considering water years with at least 80% of daily data.

Arid and semi-arid conditions prevail in the Far and Near North macrozones, with average annual precipitation of 129.1 mm and 135.2 mm (Fig. 1), respectively. In the Far North, precipitation is concentrated in the austral summer (DJF) due to the Altiplano rainy season (Garreaud et al., 2003) whereas, in the Near North, winter precipitation is driven by Cut-Off Low events that produce snow accumulation at high altitudes (Rondanelli, 2025). The Near North macrozone also exhibits the highest daily minimum (maximum) temperature values, averaging 9.1°C (24.5°C). In Central Chile, temperate climates dominate, with an
average annual precipitation of 591 mm, primarily concentrated in winter. The Southern macrozone is the wettest region, receiving an average of ~1690 mm/yr (Fig 1), with persistent winter precipitation events and moderate summer precipitation (Aceituno et al., 2021). Finally, precipitation is relatively uniform throughout the year in the Austral macrozone, with seasonal totals ranging 200-365 mm, and the lowest daily minimum (maximum) temperature values averaging 2.8°C (11.3°C).

    For hydrological analyses, we selected 44 catchments from the 2022 version of the Catchment Attributes and Meteorology
for Large Sample Studies, Chile dataset (CAMELS-CL; Alvarez-Garreton et al., 2018). The selected catchments fulfill the following criteria: (i) at least 80% coverage of daily observations during the period April/2001 – March/2018; (ii) location in





the headwaters of the main river basins of the country; (iii) less than 2% of glacierized area; (iv) absence of reservoirs; (v) area larger than 300 km$^2$; and (vi) a low degree of human intervention (< 0.05), quantified as the ratio between the mean annual flow of surface water rights (permanent continuous consumptive) and the mean annual streamflow measured in the basin outlet (Alvarez-Garreton et al., 2018). In the Far and Near North macrozones, the mean annual runoff ratio, calculated from annual runoff and annual precipitation over the basin area ($\bar{Q}/\bar{P}$) is below 0.27; in the Central and Southern macrozones, $\bar{Q}/\bar{P}$ ranges 0.09–1.12; and in the Austral macrozone, the $\bar{Q}/\bar{P}$ lies between 0.18 and 0.83.

## 3  Hydrometeorological datasets

### 3.1  Ground-based observations

In-situ measurements of precipitation and extreme temperatures were obtained from stations maintained by the DGA and the Chilean National Weather Service (DMC by its acronym in Spanish). We selected stations with at least 80% of daily records (period April/2001 – March/2018) that passed the quality control process followed by Lagos-Zúñiga et al. (2024), which includes: (i) the Buishand (1984) U Homogeneity Test for annually-averaged time series, (ii) exclusion of stations with more than two years of missing data, (iii) removal of records with minimum temperature higher than maximum temperature, and (iv) application of a quantile mapping procedure to fill missing data, selecting the best neighboring station to complete missing records following Tang et al. (2020). Besides, daily precipitation values exceeding the mean plus 2.3 times the standard deviation were removed. For daily extreme temperatures, values above the mean plus three standard deviations or below the mean minus three standard deviations were also removed (Newman et al., 2015). Hence, we considered 494 stations with precipitation records, and 97 (95) stations with minimum (maximum) temperature records for subsequent analyses.

Streamflow records were retrieved from stations maintained by the DGA, publicly available from the Center for Climate and Resilience Research (CR2) Climate Explorer (https://www.cr2.cl/datos-de-caudales/).

### 3.2  WRF-SAAG Simulations

The convection-permitting WRF-SAAG dataset (Dominguez et al., 2024) was produced by forcing the Weather Research and Forecasting Model (WRFv4.1.5, Skamarock and Klemp, 2008) with fifth-generation reanalysis (ERA5; Hersbach et al., 2020) produced by the European Centre for Medium-Range Weather Forecasts (ECMWF), using a 4-km grid spacing and 61 vertical levels over the period January/2000-December/2021. The model physics options include the Thompson microphysics scheme (Thompson et al., 2008); the Rapid Radiative Transfer Model (RRTM, Iacono et al., 2008); the Yosei University (YSU) boundary layer (Hong et al., 2006); the Noah land surface model with Multiple Parameterization Options (Noah-MP; Niu et al., 2011); and the Miguel-Macho and Fan (MMF) groundwater scheme (Miguez-Macho et al., 2007). For more details on the methodology and outcomes of these simulations, readers are referred to Dominguez et al. (2024).





### 3.3 CR2MET

The CR2MET gridded meteorological product (DGA, 2017, Boisier et al., 2018) includes daily time series with precipitation and near-surface maximum/minimum temperatures over a regular 0.05°x0.05° horizontal grid across continental Chile, for the period 1960-2021 in its latest version (v2.5). In CR2MET daily precipitation estimates were obtained through a statistical

postprocessing technique that uses topographic descriptors and large-scale climatic variables (water vapor and moisture fluxes) from the ERA5 reanalysis (Hersbach et al., 2020). The postprocessing includes (i) a logistic regression model to estimate the probability of precipitation, and (ii) multiple linear regression models to compute precipitation amounts. Maximum and minimum daily temperature were also estimated through multiple linear regression models, including variables from MODIS land surface products as additional predictors. For the first version of CR2MET, more than 800 precipitation stations were

considered (DGA, 2017). However, the number of temperature stations used in that version was not specified. Moreover, subsequent versions of the precipitation and temperature products do not provide detailed information on the number of stations used.

The CR2MET dataset has been used as an observational reference for many purposes, including the evaluation of other meteorological products (e.g., Bozkurt et al., 2019; Fernández et al., 2021; Torrez-Rodriguez et al., 2023); hydroclimatic

characterizations (e.g., Vásquez et al., 2021; Hernandez et al., 2022); the assessment of hydrological modeling decisions (e.g., Sepúlveda et al., 2022; Murillo et al., 2022; Cortés-Salazar et al., 2023); drought propagation studies (e.g., Alvarez-Garreton et al., 2021; Lema et al., 2025); streamflow forecasting (e.g., Araya et al., 2023); and climate change impact assessments (Vicuña et al., 2021; Gateño et al., 2024; Vásquez et al., 2024, 2025).

### 3.4 RF-MEP

The Random Forest based MErging Procedure (RF-MEP, Baez-Villanueva et al., 2020), uses the Random Forest algorithm to characterize the spatial distribution of precipitation by merging information from different gridded products and ground-based observations for a given temporal scale. In this study, we used daily gridded precipitation generated by Baez-Villanueva et al. (2021) for a 0.05°x0.05° horizontal grid that covers continental Chile over the period April/2001 -March/2018. The RF-MEP product used here integrates 334 rain gauges, the SRTM-v4 digital elevation model (Rabus et al., 2003), and the atmospheric

reanalysis ERA5.

The RF-MEP methodology has been replicated and evaluated for precipitation estimation in other regions of the world. Mohammed et al. (2023) applied the RF-MEP approach in the Upper Blue Nile River basin (Ethiopia) to merge ground-based measurements, satellite and reanalysis precipitation products, as well as topography-related features. The resulting merged product exhibited improved spatio-temporal representation and greater accuracy compared to individual input sources. In Chile,

Baez-Villanueva et al. (2021) applied RF-MEP and used it to evaluate its impact on the regionalisation of hydrological model parameters, showing that RF-MEP performed competitively compared to other used precipitation datasets. RF-MEP has also served as a benchmark in comparisons with other precipitation estimation methodologies. For instance, Chen et al. (2022) found that RF-MEP performed comparably to triple collocation-based fusion methods in terms of correlation and error met-



rics, while Chen et al. (2024) reported that although RF-MEP improved precipitation estimates over raw satellite data, it was
outperformed by their proposed Spatial Random Forest Downscaling and Merging (SRF-DM) approach, particularly in captur-
ing high-intensity rainfall and complex spatial patterns. Further, RF-MEP has been applied in climate change studies to assess
precipitation extremes and their trends (e.g., Valdivieso-García et al., 2024; Yan et al., 2024), and to characterize and optimize
ground-based precipitation monitoring networks (e.g., Sreeparvathy and Srinivas, 2022).

## 4    Methods

### 4.1    Evaluation of precipitation and extreme temperatures

We aggregated the WRF-SAAG hourly outputs to a daily time scale in continental Chile, starting at UTC-4 (8 hours winter
local time in Santiago). Observational products are originally provided at a daily resolution within the same time range. We
extracted, at each station location in Figure 1a, the closest grid cell values of daily precipitation time series from the WRF-
SAAG, CR2MET, and RF-MEP products. Similarly, time series with maximum (minimum) daily temperatures were retrieved
from the WRF-SAAG and CR2MET grid cells containing the stations in Figure 1c (1d), following the approach proposed
by Thiemig et al. (2012). Then, we evaluated the capability of these products to replicate the occurrence of precipitation
events with different magnitudes (section 4.1.1) at station locations and the overall skill of daily time series (section 4.1.2). It
should be noted that the assessment methods used here are very restrictive and demand that events are simulated at the exact
same time and location as observed. The WRF-SAAG simulations were designed as a climatological dataset, and might feature
spatiotemporal displacements of individual storms due to the large computational domain and chaotic nature of the atmosphere.
On the other hand, CR2MET and RF-MEP were developed using many of the station-based observations that are used in this
work for the evaluation process.

### 4.1.1    Ability to simulate observed precipitation events

We used metrics formulated from contingency tables to assess the ability of the datasets to replicate historically observed daily
precipitation events exceeding 1, 5, 10, and 20 mm. Given a precipitation threshold, 2x2 contingency tables (Table 1) can
be constructed to assess the correspondence between nonprobabilistic forecast values and the discrete observable predictand
values to which they pertain (Wilks, 2019).

We assessed the ability to replicate precipitation occurrence using the percentage correct ($PC$, ec. 1), the probability of
detection ($POD$, ec. 2), the critical success index ($CSI$, ec. 3), and the false alarm ratio ($FAR$, ec. 4) Wilks (2019). These
metrics have been widely used to assess precipitation products (e.g., Hobouchian et al., 2017; Nashwan et al., 2020; Valencia
et al., 2023). The $PC$ indicates how well the datasets discriminate precipitation and non-precipitation events:

$$PC = \frac{H + CN}{H + F + M + CN} \tag{1}$$



**Table 1.** Contingency table.

| | Event observed | |
|---|---|---|
| **Event simulated** | Yes | No |
| Yes | Hits (H) | False Alarms (F) |
| No | Misses (M) | Correct negatives (CN) |

The $POD$ evaluates the ability to capture the occurrence of events:

$$POD = \frac{H}{H + M} \tag{2}$$

We used the $CSI$ to assess how well each product captures events:

$$CSI = \frac{H}{H + F + M} \tag{3}$$

Finally, we used the $FAR$ to evaluate the fraction of events erroneously predicted:

$$FAR = \frac{F}{H + F} \tag{4}$$

$PC$, $POD$, $FAR$ and $CSI$ values range between 0 and 1, being the optimal value equal to 1 except for $FAR$, whose perfect

value is 0.

### 4.1.2 Accuracy of daily precipitation and extreme temperature simulations

To assess the accuracy of precipitation and temperature retrieved from the different products, we used a modification of the
Kling-Gupta efficiency ($KGE$; Gupta et al., 2009) proposed by Tang et al. (2021):

$$KGE_T = 1 - \sqrt{(r-1)^2 + \beta_T^2 + (\alpha_T - 1)^2} \tag{5}$$

$$\beta_T = \frac{\mu_s - \mu_o}{\sigma_o} \tag{6}$$

$$\alpha_T = \frac{\sigma_s}{\sigma_o} \tag{7}$$

where $r$, $\beta_T$ and $\alpha_T$ are the Pearson correlation coefficient, the bias and the variability term between observations ($o$) and
data set ($s$), respectively, whereas $\sigma$ and $\mu$ are the standard deviation (mm or °C) and the mean (mm or °C), respectively. Note
that $KGE_T$, $r$, $\beta_T$ and $\alpha_T$ are dimensionless, and that the optimal value for $KGE_T$, $r$, and $\alpha_T$ is 1, while the optimal value

for $\beta_T$ is 0. In this study, we computed the $KGE_T$ for precipitation by season, whereas we used the entire time series for the
assessment of temperature extremes.



## 4.2 Hydrological modeling

To assess the potential of WRF-SAAG precipitation and temperature for hydrological modeling applications and perform a comparative assessment against CR2MET and RF-MEP, we configured and calibrated the TUW model (Parajka et al., 2007) in the 44 case study basins. The TUW model is a conceptual rainfall-runoff model that simulates the catchment-scale water balance through snow, soil moisture, response, and routing modules, requiring the specification of 15 parameters (Table 2). In this study, we configure the TUW model using the TUWmodel package implemented in the statistical software R (Viglione and Parajka, 2019). To account for subunit variability in precipitation and temperature due to orographic effects, we delineated three equal-area elevation bands in catchments smaller than 600 km$^2$ and five equal-area elevation bands in the remaining basins. To this end, we used digital elevation models (DEMs) from the Shuttle Radar Topography Mission (SRTM; Rabus et al., 2003) with a 3 arc-second horizontal resolution (approximately 90 m).

The model's required input forcing variables are precipitation, temperature, and potential evapotranspiration. Hence, we extracted precipitation time series from the WRF-SAAG, CR2MET, and RF-MEP datasets, and temperature time series from the WRF-SAAG and CR2MET datasets to compare different combinations of forcing datasets: (i) WRF-SAAG precipitation and temperature; (ii) CR2MET precipitation and temperature; (iii) RF-MEP precipitation and WRF-SAAG temperature; (iv) RF-MEP precipitation and CR2MET temperature. We calculated potential evapotranspiration for each elevation band using the formulation proposed by Oudin et al. (2005), available in the R package airGR (Coron et al., 2023) and considering the latitude of the centroid of each band.

We calibrated the parameters of the TUW model for each catchment and input forcing combination by maximizing the composite metric proposed by Garcia et al. (2017):

$$OF = \frac{KGE'(Q) + KGE'(1/Q)}{2} \tag{8}$$

where $KGE'$ is the modified Kling-Gupta efficiency (KGE', Kling et al., 2012):

$$KGE' = 1 - \sqrt{(r-1)^2 + (\beta-1)^2 + (\gamma-1)^2} \tag{9}$$

$$\beta = \frac{\mu_s}{\mu_o} \tag{10}$$

$$\gamma = \frac{\sigma_s/\mu_s}{\sigma_o/\mu_o} \tag{11}$$

and $r$, $\beta$ and $\gamma$ are the Pearson correlation coefficient, the bias ratio, and the variability ratio, respectively. The values of $KGE'$, $r$, $\beta$ and $\gamma$ are dimensionless, with an optimal value of 1.

The model parameters were calibrated by maximizing the objective function in Equation 8 with the Shuffled Complex Evolution (SCE-UA, Duan et al., 1993) global optimization algorithm, using the same parameter ranges as Araya et al. (2023, see Table 2). The calibration period was defined as April/2005 – March/2011, allowing a 4-year warm-up period, and the evaluation period was set as April/2011 – March/2018.





**Table 2.** Description of TUW model parameters and calibration ranges.

| Parameter | Description | Units | Range |
|-----------|-------------|-------|-------|
| SCF | Snow correction factor | – | $0.5 - 2$ |
| DDF | Degree day factor | mm/°C/day | $0 - 5$ |
| Tr | Threshold temperature above which precipitation is rain | °C | $1 - 5$ |
| Ts | Threshold temperature below which precipitation is snow | °C | $-3 - 1$ |
| Tm | Threshold temperature above which melt starts | °C | $-2 - 4$ |
| Lprat | Parameter related to the limit for potential evaporation | – | $0 - 1$ |
| FC | Field capacity | mm | $0 - 1000$ |
| BETA | Nonlinear parameter for runoff production | – | $0 - 20$ |
| k0 | Storage coefficient for very fast response | day | $0 - 2$ |
| k1 | Storage coefficient for fast response | day | $2 - 30$ |
| k2 | Storage coefficient for slow response | day | $30 - 500$ |
| LSuz | Threshold storage state | mm | $1 - 100$ |
| Cperc | Constant percolation rate | mm/day | $0 - 10$ |
| Bmax | Maximum base at low flows | day | $0 - 30$ |
| Croute | Free scaling parameter | day$^2$/mm | $0 - 50$ |

# 5  Results

## 5.1  Precipitation and temperature evaluation

### 5.1.1  Ability to simulate precipitation events

Figure 2 shows categorical metrics that assess the ability of WRF-SAAG, CR2MET, and RF-MEP to replicate precipitation events of different magnitudes across the study macrozones. In general, all datasets provide high values of percentage correct ($PC \geq 0.64$) in all stations and for all precipitation thresholds, influenced by the number of correct negatives ($CN$). The results for $POD$ – which only considers correctly detected precipitation events ($H$) – show that, overall, the ability of WRF-SAAG and CR2MET to replicate precipitation events decreases with higher precipitation thresholds. In general, RF-MEP achieves the highest $POD$ values, particularly in Central Chile ($POD_{RF-MEP} \geq 0.65$). For events $> 1$ mm/d, CR2MET achieves the highest POD values across all study zones, with $POD_{CR2MET}$ medians $\geq 0.85$.

Figure 2 also shows that RF-MEP maintains the highest $CSI$ values between the Near North and Southern macrozones. In the Far North, all the datasets fail to replicate the fraction of observed precipitation events (i.e., $CSI \leq 0.5$ in most stations), in agreement with the high false alarm ratio ($FAR$) in this macrozone, especially for events exceeding 20 mm/d (median $FAR \geq 0.86$). Similarly, WRF-SAAG and RF-MEP yield large $FAR$ ranges in the Austral macrozone ($IQR$ of $FAR \geq 0.23$),



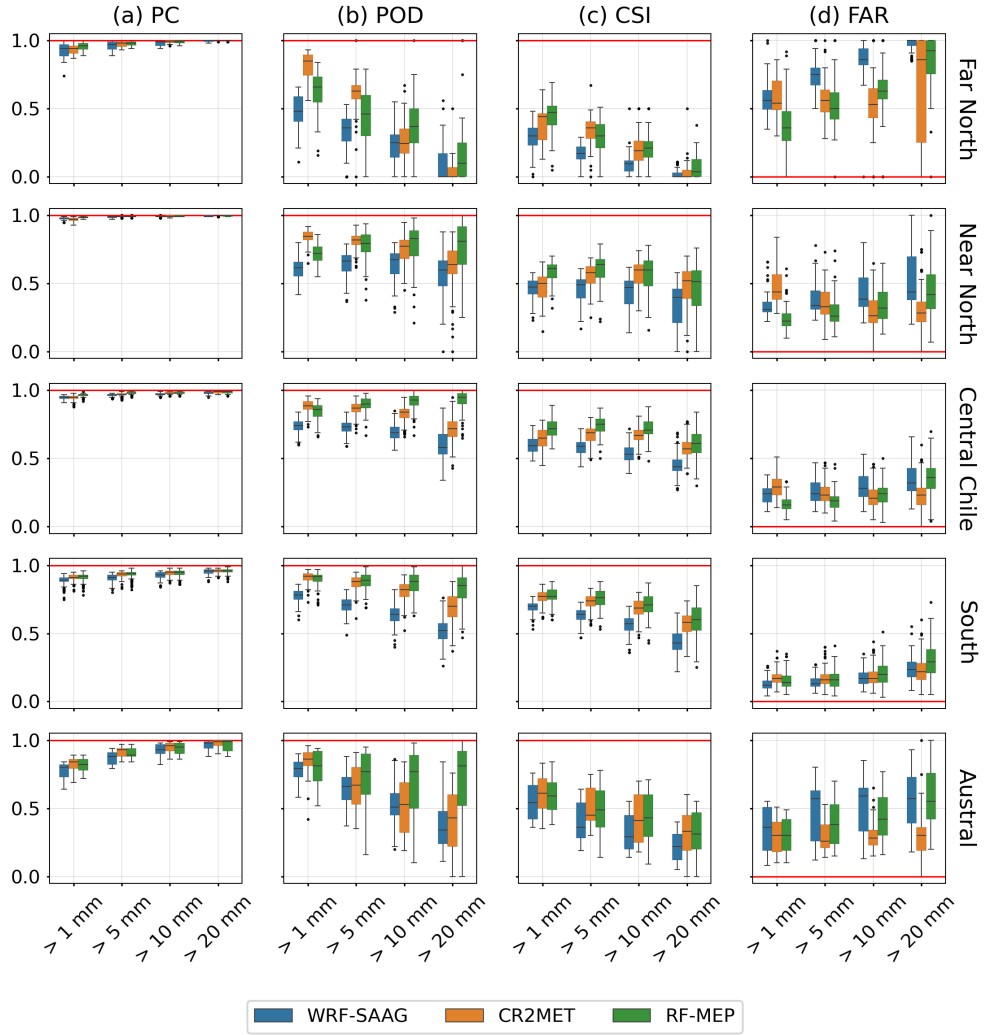

**Figure 2.** Ability of WRF-SAAG, CR2MET, and RF-MEP to replicate historically observed daily precipitation events with amounts larger or equal than 1, 5, 10, and 20 mm/d in the study macrozones (rows). The metrics displayed in each column are: (a) Percentage Correct ($PC$), (b) Probability of detection ($POD$), (b) Critical Success Index ($CSI$), and (d) False Alarm Ratio ($FAR$). Each boxplot comprises results from all the stations within each macrozone. The boxes correspond to the interquartile range ($IQR$, i.e., 25th and 75th percentiles), the horizontal line in each box is the median, and whiskers extend to the $\pm 1.5 \cdot IQR$ of the ensemble. The red line represents the optimal values.

in contrast to CR2MET ($IQR$ of $FAR \leq 0.22$). Moreover, for precipitation events above 5 mm/d, all datasets provide median $CSI$ values below 0.49 in the Austral macrozone. In contrast, Central Chile shows better detection performance, with a median $CSI \geq 0.44$ for events above 5 mm/d.





### 5.1.2 Accuracy of daily precipitation estimates

Figure 3 shows the spatial distribution of $KGE_T$ (and its components) for summer (DJF) daily precipitation estimates from WRF-SAAG, CR2MET, and RF-MEP. The highest $KGE_T$ values in summer are obtained in the Southern macrozone, where $KGE_T$ medians $\geq 0.65$ for all datasets and seasons. The second best $KGE_T$ for CR2MET is obtained in the Austral region (median $KGE_T = 0.72$), whereas the second best $KGE_T$ for WRF-SAAG (RF-MEP) is achieved in Central Chile, with a median of 0.59 (0.57). However, there are also negative $KGE_T$ values within this macrozone, near its upper boundary

($\sim 32.18°$S).

    The results in Figure 3 show that the datasets have limitations in effectively capturing daily precipitation in arid regions, characterized by very low (or zero) precipitation amounts. In the Far North and Near North, 23 stations did not record any precipitation events during summer, which yields undefined $KGE_T$ at these sites. In spite of this, the number of precipitation events exceeding 1 mm/d at the corresponding grid cell were 232, 484 and 18 for WRF-SAAG, CR2MET and RF-MEP,

respectively. The lowest summer $KGE_T$ values are obtained in the Near North, with medians $\leq -0.18$ for all datasets. In this subdomain, a cumulative summer average of only 1 mm is recorded by the stations. Additionally, the largest differences among datasets in terms of $KGE_T$ are attained in the Far North and Austral macrozones, with CR2MET standing out in both regions, with medians of 0.57 and 0.72, respectively.

    The largest spread among the $KGE_T$ components for the three datasets is obtained for the variability ratio ($\alpha_T$; Figure 3f),

making it the term with the largest influence on the spatial variability of $KGE_T$ throughout all the seasons (see also Figures 4, A1, and A2). Furthermore, WRF-SAAG provides lower correlation coefficients ($r$) compared to CR2MET and RF-MEP in all macrozones (Figure 3d), which is consistent with the results obtained for the remaining seasons. Figure 3e shows that RF-MEP tends to overestimate daily precipitation amounts (median $\beta_{T_{RF-MEP}} \geq 0.03$) and overestimate precipitation variability (median $\alpha_{T_{RF-MEP}} \geq 1.12$), a behavior that persists throughout the rest of the year, except for autumn and winter in the Far

North (the results for the transition seasons are provided in the appendix). Additionally, CR2MET represents precipitation amounts between the Central and Austral macrozones better than the other datasets throughout the year (median $\beta_T$ difference of 0.03 from the optimal value; Figure 3e). However, CR2MET underestimates the variability of daily precipitation during summer in this area, with $\alpha_{T_{CR2MET}}$ medians $\leq 0.87$. This behavior is also observed in the remaining seasons.

    Figure 4 displays performance metrics for winter (JJA) daily precipitation across the study domain. In general, both WRF-

SAAG and CR2MET show an increase in median $KGE_T$ during winter compared to summer between the Near North and Austral macrozones. CR2MET achieves the highest median winter $KGE_T$ in all macrozones, with the best performance in the Central and Southern macrozones (median $KGE_T \geq 0.8$). In the Near North, the three datasets shift $KGE_T$ from negative values in summer to medians above 0.61 during winter. The Far North has the lowest winter precipitation (less than 5 mm per station) and the lowest median $KGE_T$ compared to the other macrozones, with median values of -0.14, 0.47 and 0.19

for WRF-SAAG, CR2MET, and RF-MEP, respectively. The best results for WRF-SAAG (RF-MEP) are obtained in Central (Southern) Chile, with median $KGE_T = 0.76$ (0.77). On the other hand, RF-MEP performs better during summer than in



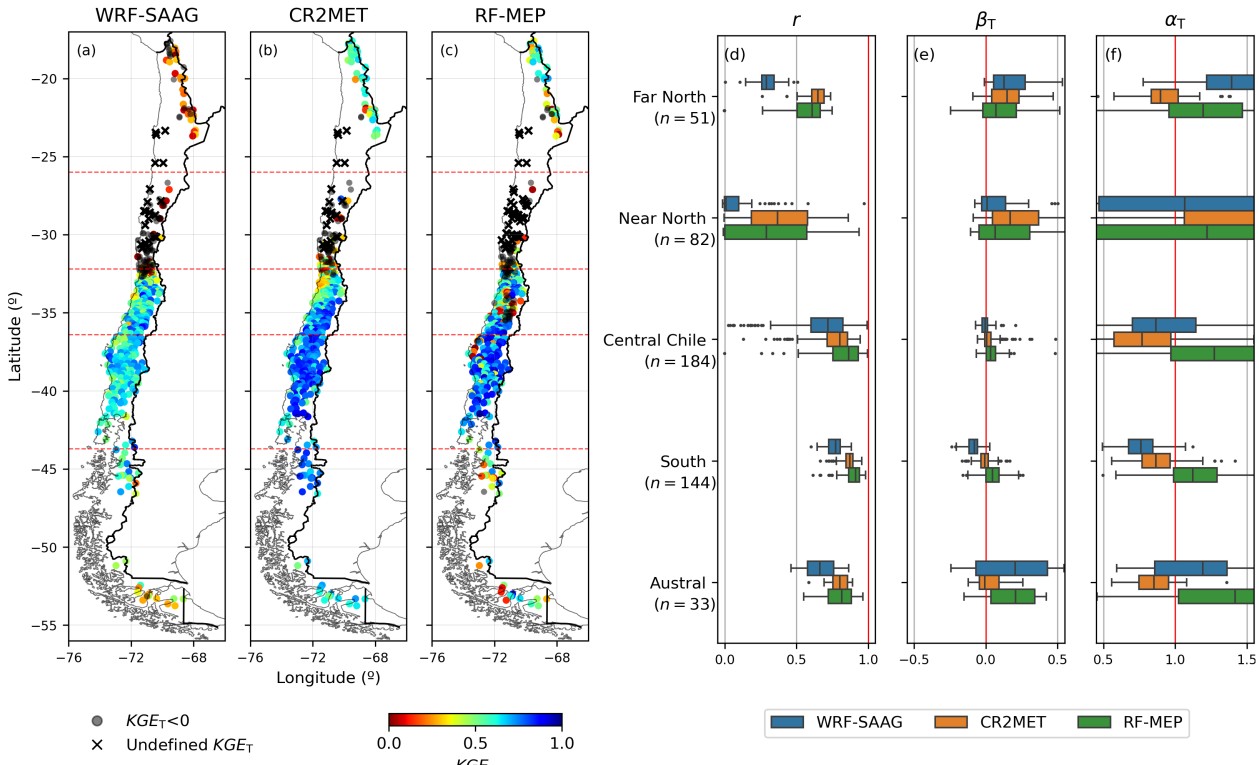

**Figure 3.** (a)–(c) Spatial distribution of Tang's Kling-Gupta Efficiencies ($KGE_T$) for summer (DJF) daily precipitation retrieved from WRG-SAAG, CR2MET, and RF-MEP (period April/2001–March/2018) across continental Chile, using ground-based observations as the reference. Red dashed lines indicate the limits of the macrozones analyzed. (d) to (f) Boxplots with the Pearson correlation coefficients ($r$), bias term ($\beta_T$), and variability ratios of Tang's Kling-Gupta Efficiency ($\alpha_T$) obtained with the products examined for each macrozone, with the red line indicating the optimal value. Each boxplot comprises results for all the stations within a specific macrozone. The boxes correspond to the interquartile range (IQR, i.e., 25$^{th}$ and 75$^{th}$ percentiles); the vertical line in each box is the median, and the whiskers extend to the $\pm 1.5 \cdot IQR$ of the ensemble of stations.

winter in the Central ($KGE_T$ median: 0.57 and 0.5 respectively) and Austral ($KGE_T$ median: 0.46 and 0.44, respectively) macrozones.

During winter, the three datasets yield the lowest Pearson correlation in the Far North (Figure 4d), with medians $\leq 0.63$, and
large ranges in the variability term $\alpha_T$ (Figure 4f). Interestingly, CR2MET achieves a median $\alpha_T$ equal to the optimal value, but with an interquartile range of 0.51.

Figure 4 shows that CR2MET excels in replicating winter precipitation volumes, with bias term medians deviating by only 0.05 from the optimal value. Further, CR2MET tends to underestimate daily precipitation variability in these regions, with $\alpha_T$ medians $\leq 0.92$. On the other hand, RF-MEP struggles to replicate precipitation variability, overestimating the dispersion of
precipitation amounts between the Near North and Austral macrozones (i.e., $\alpha_T$ medians spanning 1.1 - 1.47).





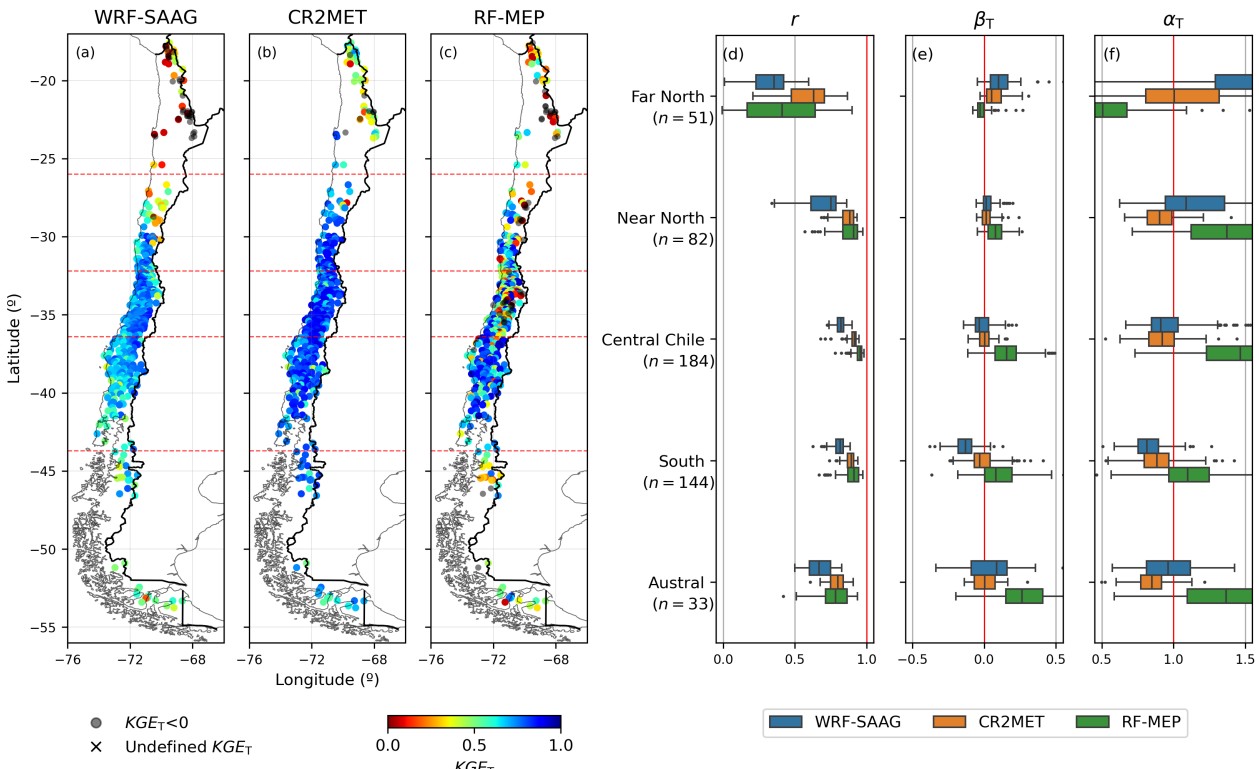

**Figure 4.** Same as in Figure 3, but for winter (JJA) daily precipitation.

### 5.1.3 Accuracy of daily temperature estimates

Figure 5 displays the spatial distribution of $KGE_T$ and its components for WRF-SAAG and CR2MET minimum daily temperatures. The results show that CR2MET outperforms WRF-SAAG, with median $KGE_T \geq 0.71$ in the Central, Southern and Austral macrozones, while the median values drop between 0.52 and 0.54 in the northern regions. WRF-SAAG reaches
its highest $KGE_T$ values in the Austral macrozone, with a median of 0.6; on the other hand, the median $KGE_T$ between the Far North and Central Chile macrozones is $\leq$0.43. Figure 5c also shows that CR2MET yields a higher Pearson correlation in all macrozones, with median values $\geq 0.74$, and a narrower interquartile range of $\beta_T$ and $\alpha_T$ coefficients compared to WRF-SAAG.

The results in Fig. 5d show that CR2MET tends to underestimate minimum temperatures (median $\beta_T < 0$), with smaller
biases from the Central macrozone southward, reaching median $\beta_T$ values between -0.04 and -0.11. Conversely, WRF-SAAG overestimates minimum temperature magnitudes between the Far North and Central Chile macrozones, with median $\beta_T$ values ranging from 0.2 to 0.51. WRF-SAAG underestimates temperatures in the southern regions, reaching its best result (median $\beta_T$ of -0.13) in the Southern macrozone. The results show that CR2MET underestimates the variability of minimum temperature





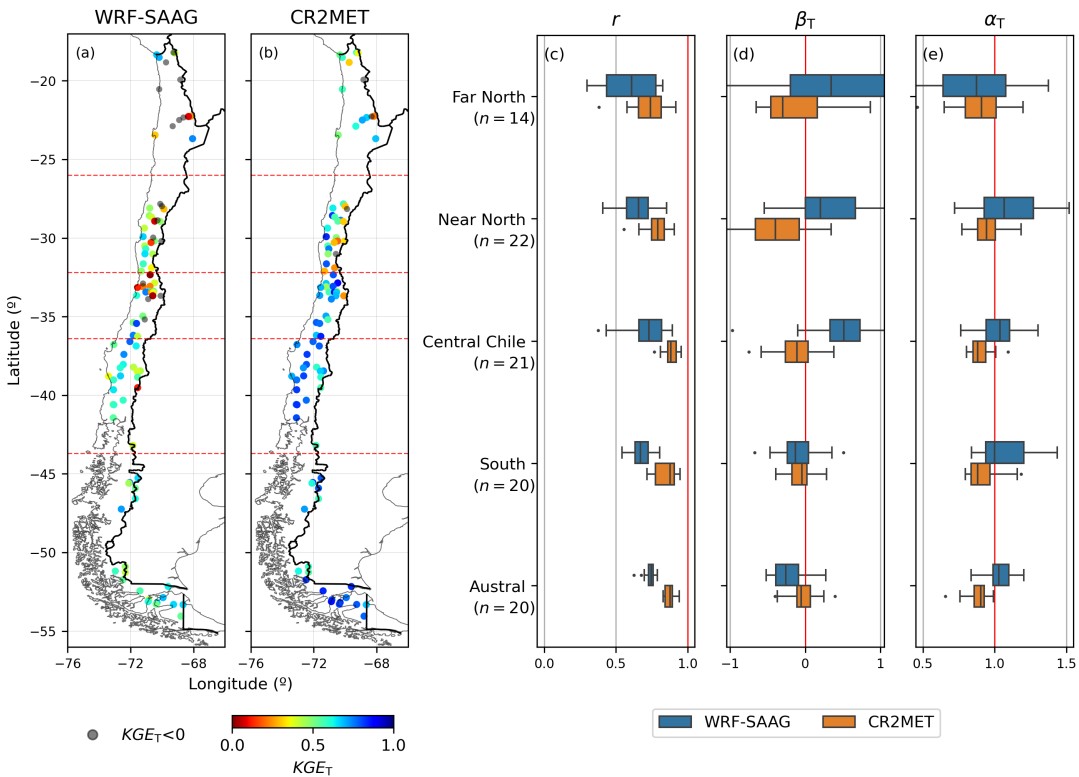

**Figure 5.** Spatial distribution of Tang's Kling-Gupta Efficiencies ($KGE_T$) for daily minimum temperature retrieved from (a) WRG-SAAG and (b) CR2MET (period April/2001 – March/2018) across continental Chile, using ground-based observations as the reference. Red dashed lines indicate the limits of the macrozones analyzed. Boxplots with (c) the Pearson correlation coefficients ($r$), (d) bias term ($\beta_T$), and (e) variability ratios ($\alpha_T$) obtained with the products examined for each macrozone, with the red line indicating the optimal value. Each boxplot comprises results for all the stations within a specific macrozone. The boxes correspond to the interquartile range ($IQR$, i.e., 25th and 75th percentiles); the vertical line in each box is the median, and the whiskers extend to the $\pm 1.5 \cdot IQR$ of the ensemble of stations.

(Fig. 5e), with median $\alpha_T$ values ranging 0.88 - 0.94 across all macrozones. In contrast, WRF-SAAG achieves median $\alpha_T$

values closer to the optimal range (1 - 1.04), particularly in the Central and Austral macrozones.

CR2MET provides higher $KGE_T$ for maximum temperature (Figure A3), compared to minimum temperature, in all macrozones except the Near North. Similarly, WRF-SAAG yields higher $KGE_T$ for maximum temperature in the Far North, Central Chile and Southern macrozones. Both CR2MET and WRF-SAAG yield general improvements in Pearson correlation for maximum daily temperature compared to minimum temperature, with median values ≥0.84 across the study area, and a reduced

range of $\alpha_T$. CR2MET tends to overestimate maximum temperatures in the Far North, Central Chile, and Southern macrozones, with median $\beta_T$ values ranging 0.01 - 0.11. In the Near North, CR2MET yields a median $\beta_T = -0.61$. Conversely, WRF-SAAG consistently underestimates maximum temperatures across the study domain (median $\beta_T \leq -0.05$), with the lowest median $\beta_T$ values in the Far North (-0.63) and Near North (-0.6) macrozones, respectively.





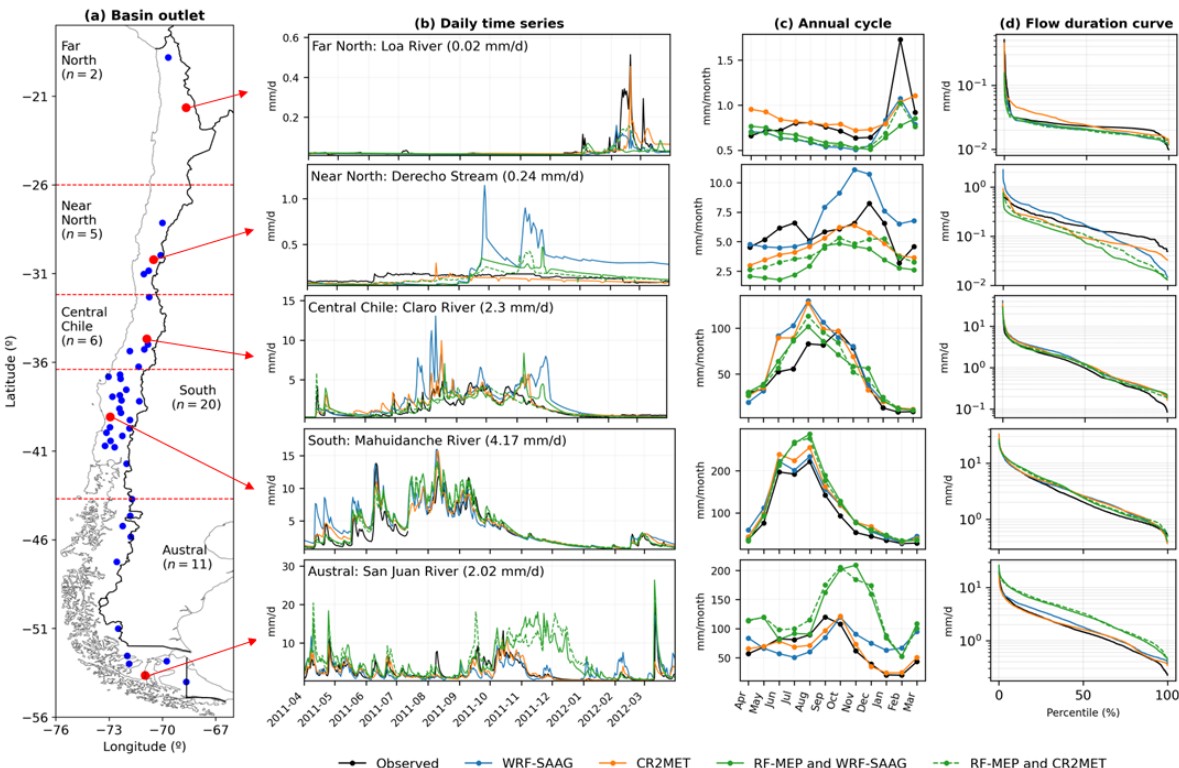

**Figure 6.** (a) Location of the 44 stream gauges and five (5) selected catchment outlets (one per macrozone, red dots) used to illustrate how the choice of forcing input combination affects runoff simulations (panels b–d). (b) Time series with observed and simulated daily runoff for the period April/2011–March/2012, using different meteorological forcing inputs (the value in parentheses indicates the mean daily runoff for the study period, April/2001 – March/2018); (c) annual cycles, and (d) daily flow duration curves for the evaluation period (April/2001 – March/2018). The selected catchments are: (Far North) Loa River; (Near North) Derecho Stream; (Central Chile) Claro River; (South) Mahuidanche River; (Austral) San Juan River at the mouth.

## 5.2 Hydrological model performance

Figure 6 illustrates hydrological modeling results for five (5) selected river basins (one for each macrozone). In general, we obtained that the parameter estimation process has a greater capability to compensate for differences in input forcings over domains with temperate climates (i.e., Central and Southern Chile). Figures 6b and 6c show that all input forcing combinations fail in simulating annual runoff cycles at the Loa River basin (Far North) and the Estero Derecho (Near North). In the Loa River basin, none of the configurations is able to capture the flood events observed in February 2012. All the forcing combinations

fail to reproduce the winter runoff increase (JJA) and low flow volumes in the Estero Derecho basin; further, a flashier response (i.e., steeper slope of the mid-segment of flow duration curve) is obtained with all forcing datasets.





Between the Central Chile and Austral macrozones, the mean daily runoff across stations ranges from 0.2 to 9.5 mm/d. In the Claro River (Figure 6, Central Chile macrozone), all the forcing combinations episodically overestimate runoff volumes during the peak runoff season (July–September), with differences between simulated and observed mean monthly runoff ranging from 4.2 to 47.4 mm/month, and slight overestimations of low flows (lower percentiles in the flow duration curve, Figure 6c). In the Mahuidanche River basin (Southern macrozone), all the forcing combinations capture annual cycles and flow duration curves, but RF-MEP-based datasets tend to overestimate daily runoff. In the San Juan River basin (Austral macrozone), RF-MEP configurations yield considerably larger runoff volumes, with differences in mean monthly runoff between simulated and observed values exceeding 120 mm/month between November and December, and consistent overestimations in the flow duration curve.

Figure 7 compares hydrologic model calibration and evaluation results obtained with different forcing combinations. Overall, the results reveal comparable performance metrics between WRF-SAAG and CR2MET. In the Far North, $OF \leq 0.61$ for all model configurations over the calibration period, and $OF \leq 0.44$ during the evaluation period, where the models also underestimated the variability of observed runoff ($\gamma \leq 0.69$). In this macrozone, the Pearson correlation coefficient ($r$) ranged 0.33 - 0.76 during the calibration period, whereas in the evaluation period, $r \leq 0.47$ in 7 out of 8 cases (4 model combinations for both basins). In the Near North, the $OF$ ranged 0.48 - 0.87 during calibration for all forcing combinations, while in the evaluation period $OF \leq 0.45$; calibration $\beta$ and $\gamma$ values varied between 0.91 and 1.05 (i.e., little spread among catchments) whereas, in the evaluation period, the range of $\beta$ (i.e., difference between $\beta_{max}$ and $\beta_{min}$ among five basins within the macrozone) increased to values $\geq 0.96$, and the range of $\gamma$ also expanded to $\geq 0.75$.

In Central Chile, the median $OF \geq 0.83$ and median $r$ ranged 0.84 – 0.86 during the calibration period for all forcing combinations, decreasing to $OF \geq 0.58$ and $r$ values of 0.71 - 0.74 in the evaluation period. Further, an increase in the interquartile range ($IQR$) of $\beta$ ($\gamma$) was obtained, with values of $IQR \leq 0.03$ ($IQR \leq 0.07$) during calibration, expanding to $IQR \geq 0.23$ ($IQR \geq 0.28$) during the evaluation period.

In the Southern macrozone and for all model configurations, no significant differences were found between calibration and evaluation periods ($p > 0.05$ obtained from a Student's t-test) for $r$, $\beta$, and $\gamma$, except for $\beta$ in configurations including RF-MEP. The runoff simulations with the RF-MEP precipitation product struggle to represent runoff volumes ($IQR \geq 0.25$ for $\beta$). Regarding the $OF$, the median remains above 0.87 (0.67) during calibration (evaluation), with WRF-SAAG and CR2MET products providing greater consistency ($IQR \leq 0.31$) during both periods. In the Austral macrozone, the simulations with RF-MEP precipitation data yield an overestimation of runoff volumes during the evaluation period (median $\beta \geq 1.12$).



**Figure 7.** Hydrologic model calibration and evaluation results for the 44 case study basins using different forcing input combinations: WRF-SAAG (precipitation and temperature); CR2MET (precipitation and temperature); RF-MEP (precipitation) and WRF-SAAG (temperature); RF-MEP (precipitation) and CR2MET (temperature). The columns indicate the following metrics: (a) objective function (OF) used for calibration, (b) Pearson correlation coefficient ($r$), (c) bias ratio ($\beta$), and (d) variability ratio ($\gamma$). Each boxplot comprises results for all the stations within a specific macrozone (displayed in different rows). The boxes correspond to the interquartile range (IQR, i.e., $25^{th}$ and $75^{th}$ percentiles); the vertical line in each box is the median, and the whiskers extend to the $\pm 1.5 \cdot$ IQR of the ensemble of stations. Points are used instead of boxplots for the Far North macrozone (first row) because only two basins were evaluated. The red line represents the optimal value.





## 6   Discussion


WRF-SAAG simulations struggled to replicate observed daily precipitation events of different magnitudes, compared to CR2MET and RF-MEP, which integrate ground observations in their development. This result is somewhat expected, since the evaluation dataset is fully independent of WRF-SAAG, as opposed to CR2MET and RF-MEP. Additionally, this result aligns well with Zambrano-Bigiarini et al. (2017), who showed that station-corrected datasets performed considerably better

in estimating daily precipitation events in Chile. The largest discrepancies among the datasets were observed in the Far North macrozone, where convective precipitation – characterized by intense and localized rainfall in the Altiplano region (Garreaud, 2000) – poses a considerable challenge for replicating such events. However, it is important to note that WRF-SAAG is a free-running climate simulation that was not designed to reproduce specific events, but rather to capture long-term statistical characteristics of hydroclimatic variables, such as mean values or variability across South America. As such, the evaluation

conducted here—based on event-scale comparisons at point locations—represents a particularly stringent benchmark. Given these constraints, the ability of WRF-SAAG to approximate observed precipitation patterns across diverse climatic regions is noteworthy, and highlights its potential value for long-term climate change studies in Chile. On the other hand, RF-MEP stood out for its ability to replicate historically observed precipitation events > 5 mm between the Near North and Southern macro-zones, with comparatively higher $POD$ and $CSI$ values than the other two datasets. This is consistent with the well-known

ability of the RF technique to enhance precision in detecting daily precipitation by incorporating topographic information, particularly in regions with complex terrain (Mohammed et al., 2023).

Daily precipitation amounts are generally better simulated by the three datasets during winter in Central and Southern macrozones mainly due to the stronger influence of synoptic scale forcing and the associated more frequent and larger scale precipitation events. However, large discrepancies arise in the northernmost and southernmost regions. Modeling daily precip-

itation in Northern Chile during summer is particularly challenging due to the difficulty of representing precipitation in arid and semi-arid zones (Cattani et al., 2016; Dinku et al., 2011). Previous studies have reported similar problems with reanalysis and satellite-based gridded products over this region, regardless of the station corrections (e.g., Schumacher et al., 2020b).

WRF-SAAG simulations showed limitations in replicating daily precipitation amounts, which could be improved through the application of statistical post-processing techniques (e.g., Mendoza et al., 2015; Meech et al., 2020). However, recent work by

Liu et al. (2025) demonstrated the robust performance of convection-permitting WRF model simulations in replicating observed climatic patterns across South America, including the spatial distribution of annual, seasonal, and sub-seasonal precipitation amounts. They also showed the ability of WRF-SAAG to capture sharp near-surface temperature gradients over steep and narrow mountains, along with the probability distributions of daily and hourly temperatures, highlighting the capability in resolving fine-scale climatic variability. One considerable advantage of WRF-SAAG is that it reproduces all atmospheric and

land surface variables with physical consistency, making it a powerful resource for complex process-based hydrological models, which require not only precipitation and temperature but also wind speed, humidity, air pressure, and radiative fluxes.

The results presented here show that CR2MET yields better performance than WRF-SAAG at the station level, which is an expected result since CR2MET combines reanalysis and ground-based observations. In areas with lower station density, ERA5



gains greater relevance in the generation of the CR2MET and RF-MEP products. Schumacher et al. (2020a) compared high-
resolution WRF precipitation simulations against ERA-Interim (ERA-I; Dee et al., 2011) reanalysis estimates in the Andean
sector of central Chile. They found that WRF outputs outperform ERA-I in representing precipitation, latitudinal gradients,
seasonal variability, and extreme events in the central Andes due to its higher horizontal resolution, which enables capturing
orographic effects, particularly at elevations above 1300 m a.s.l.

The evaluation of daily extreme temperature estimates revealed that meteorological products generally exhibit better per-
formance for maximum temperature than for minimum, in agreement with previous studies (e.g., Schubert and Henderson-
Sellers, 1997; Kostopoulou et al., 2007). This is primarily because minimum temperature typically occurs at dawn, when the
atmosphere is most vertically stratified. When synoptic-scale atmospheric mixing is weak, minimum temperature tends to be
spatially heterogeneous. Moreover, a point measurement may not be locally representative in regions with complex topography
(Thorne et al., 2016).

The hydrological modeling results showed that WRF-SAAG produced similar streamflow performance metrics compared to
CR2MET, which was originally conceived for water balance calculations under historical and future climate scenarios (e.g.,
DGA, 2017, 2018, 2019) and has been subsequently used in several studies involving hydrological modeling (e.g., Vásquez
et al., 2021; Murillo et al., 2022; Sepúlveda et al., 2022; Araya et al., 2023; Cortés-Salazar et al., 2023; Muñoz-Castro et al.,
2023; Lema et al., 2025). Further, RF-MEP precipitation time series provided generally worse performance metrics. The Far
and Near North macrozones remain a challenge for hydrological modeling due to low observed precipitation amounts and
low runoff. It is also worth noting that the use of a simple bucket-style rainfall–runoff model, while advantageous for broad-
scale application across many basins, may overlook key hydrological processes (Minville et al., 2014). The simplifications
involved may particularly affect regions like the Far North, where subsurface and groundwater flows dominate (Magaritz et al.,
1990), and Near North and Central regions, where snow accumulation and melt play a critical role (Stehr and Aguayo, 2017).
Therefore, the reduced model performance obtained in some catchments cannot be solely attributed to meteorological inputs,
and additional work is required to examine other sources of uncertainty.

Although this study provides a comprehensive evaluation of WRF-SAAG precipitation and temperature fields over conti-
nental Chile, several limitations and opportunities for future work remain. First, the calibration and evaluation of hydrological
simulations could benefit from incorporating a broader range of water years, especially given the influence of the persistent
Mega-drought in Chile from 2010 to 2018 (Garreaud et al., 2020). This is particularly important considering that the transfer
of hydrological model parameters between calibration and evaluation periods with contrasting climatic conditions can lead to
performance losses (Coron et al., 2012). Second, the scope of evaluation can be expanded beyond streamflow, incorporating
other hydrological variables such as evapotranspiration, soil moisture, or snow water equivalent (SWE), derived from simu-
lated outputs from the Noah-MP land surface model included in the WRF-SAAG experiments. Finally, although beyond the
scope of the current work, the WRF-SAAG dataset could also serve as a baseline for future climate change impact studies. For
instance, bias correction using high-resolution datasets like CR2MET would enable the evaluation of pseudo-global warming
(PGW) experiments to assess future hydrological conditions under climate change scenarios.



# 7    Conclusions

We evaluated the performance of two widely used gridded meteorological datasets in Chile - CR2MET and RF-MEP -, along-
side the recently released high-resolution convection-permitting regional simulations from the Weather Research and Forecast-
ing for South America (WRF-SAAG). First, we assessed daily precipitation and extreme temperature estimates from the three
datasets against ground-based observations, with particular attention to their ability to replicate historically observed precipi-
tation events. We also quantified their accuracy with a modified version of the Kling-Gupta Efficiency. Secondly, we assessed
the suitability of these products for hydrological modeling by calibrating the parameters of a conceptual, bucket-style rainfall
runoff model in a suite of 44 hydroclimatically diverse catchments.

The main outcome of this work is that, despite not being designed to replicate individual precipitation events in space
and time, WRF-SAAG delivered comparable performance to observation-based products, especially in hydrological model
simulations. Further, WRF-SAAG provides a vast number of physically-consistent atmospheric and surface variables, making
it a promising resource for hydrological studies in regions with complex topography and sparse ground observations. Future
research may benefit from exploring post-processing techniques to further enhance the representation of precipitation and
temperature of km-scale convection permitting models. Additionally, the results obtained here demonstrate the superiority of
observation-based products - especially CR2MET - if the end goal is to examine precipitation extremes and temperature at point
locations. Finally, the relatively poorer performance of the three datasets in arid domains and data scarce regions highlights
areas of action for further research to improve gridded meteorological products.

*Data availability.*    The CR2METv2.5 dataset is available at https://www.cr2.cl/datos-productos-grillados/ (Boisier et al., 2018). The RF-
MEP dataset used in this study was developed by Baez-Villanueva et al. (2021) and is described in detail in their publication. It was generated
using the RFmerge R package by combining daily records from 334 rain gauges (available at http://www.cr2.cl/datos-de-precipitacion/,
last access: 10 January 2021), ERA5 reanalysis data, and SRTMv4.1 elevation data. The dataset covers continental Chile at a horizontal
resolution of 0.05° for the period 1990–2018. The WRF-SAAG climate simulations used in this study are available at https://ral.ucar.edu/
projects/south-america-affinity-group-saag/model-output.

*Author contributions.*    SS, PM, and ML conceptualized the study, designed the overall approach, and wrote the manuscript. SS conducted
all the evaluations of meteorological datasets, the calibration and evaluation of the hydrological model, performed the data analyses, and
produced all the figures. LS provided assistance for processing the WRF-SAAG climate simulations. AP provided support in setting up the
scripts used in this study. All authors contributed to refining the methodology and analysis framework, interpreting the results, and reviewing
and editing the manuscript.

*Competing interests.*    The authors declare that they have no conflict of interest.





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



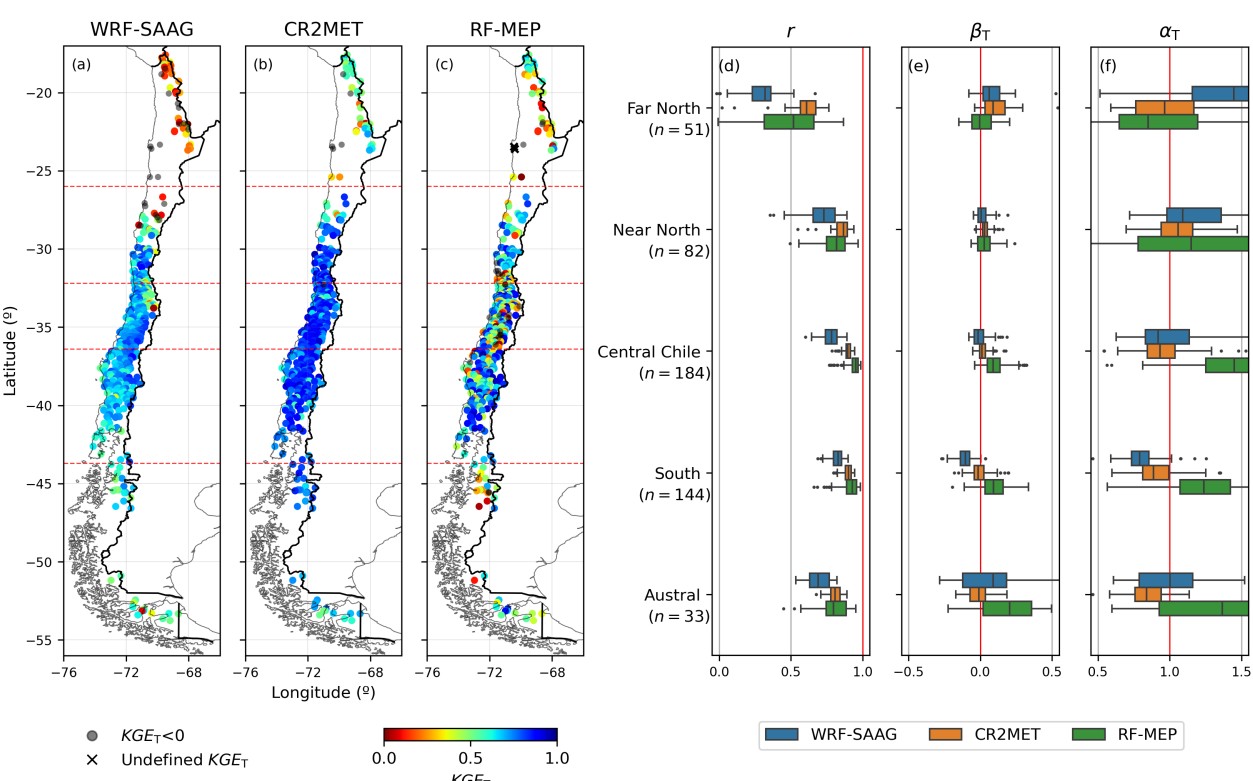

**Figure A1.** Same as in Figure 3, but for autumn (MAM) daily precipitation.



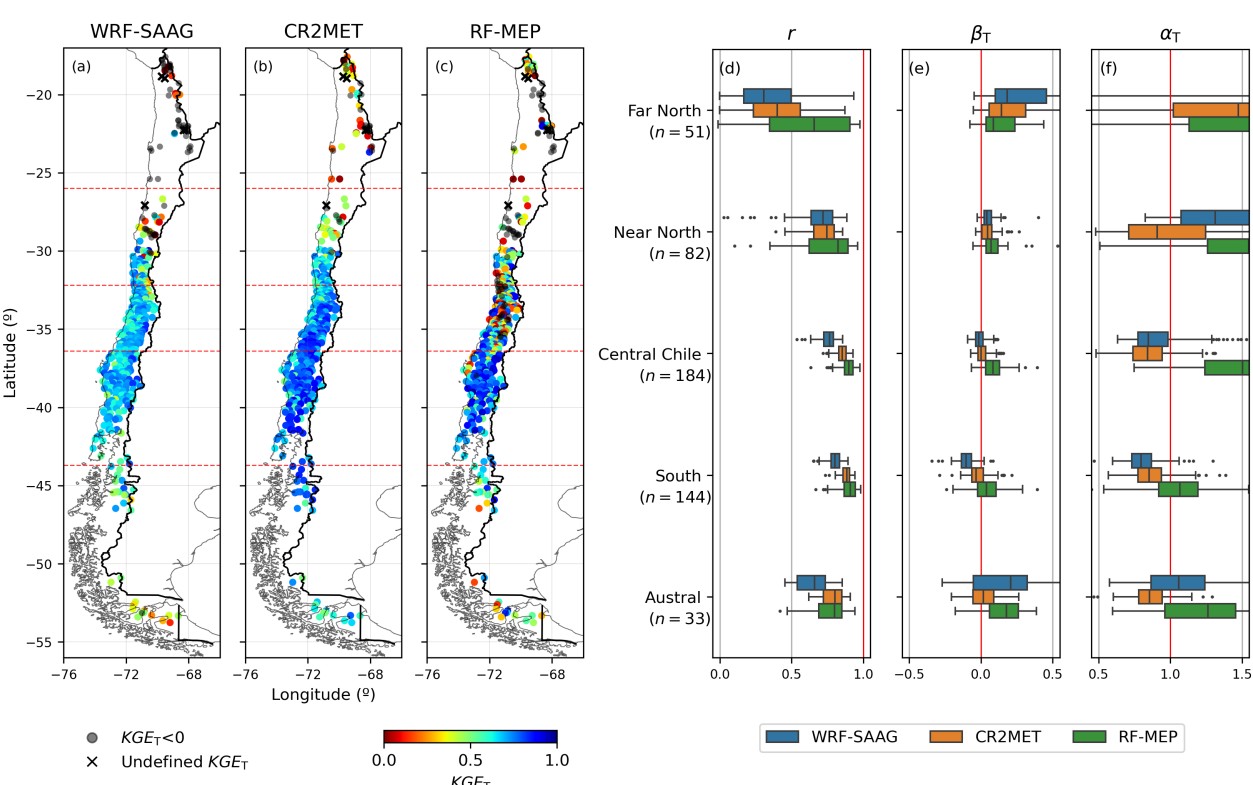

**Figure A2.** Same as in Figure 3, but for spring (SON) daily precipitation.



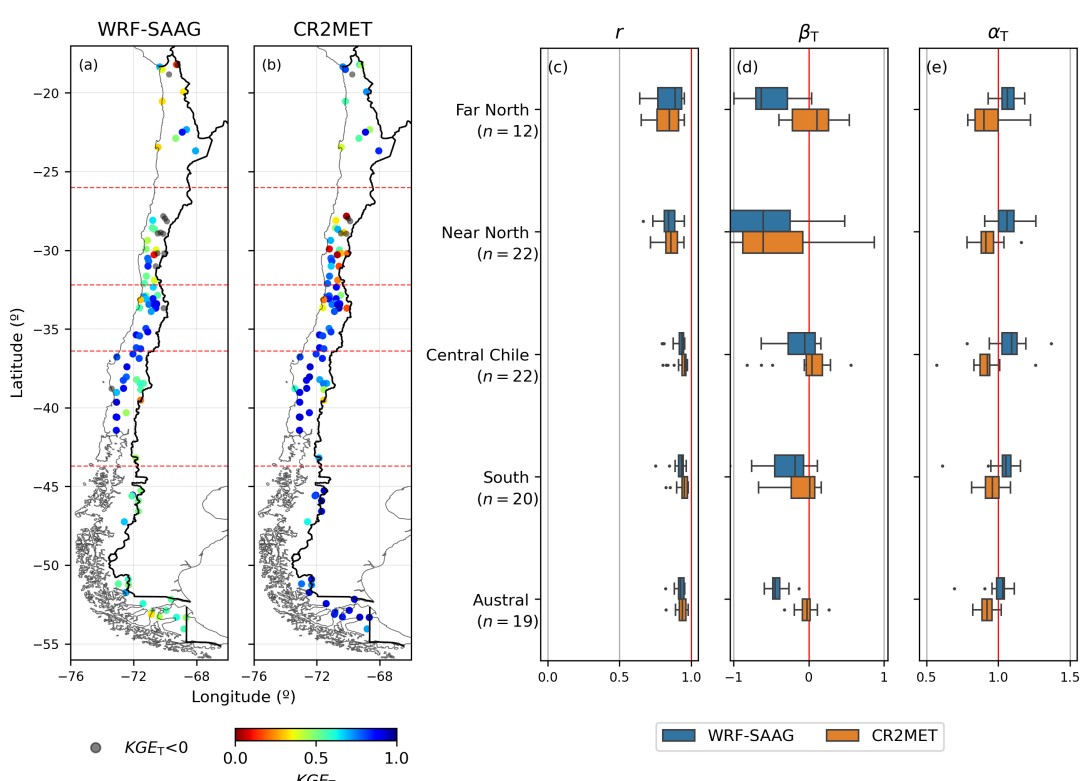

**Figure A3.** Same as in Figure 5, but for maximum temperature.