# Peer review of "Benchmarking convection-permitting climate simulations for hydrological applications: A comparative study of WRF-SAAG and observation-based products"

_EGUsphere, 2025_

## Referee Comment (RC1)

**GENERAL COMMENTS**

The manuscript offers an evaluation of daily total precipitation, maximum daily temperature, and minimum daily temperature variables simulated by the WRF-SAAG convective-permitting numerical model for hydrological applications. It compares these outputs with meteorological station measurements and simulations from the CR2MET and RF-MET products in the conterminous Chile. Furthermore, these variables are used as input data in a conceptual hydrological model (HBV-like) in order to assess the performance of the simulated daily streamflow against observed series in pristine catchments with low glacier cover.

The work addresses a fundamental topic in mountain regions, which is the use of numerical models to supplement the lack of meteorological measurements, especially for precipitation. However, the article has certain **structural deficiencies** that require manuscript **major revisions**,

1. The title and the body of the text do not specify which hydrological applications are being adressed to (e.g., flood risk, rain-on-snow events, water supply projections, among others - see a non-exhaustive list in Table 1 of Dominguez et al., 2024). The associated temporal scale and hydrological processes are also not defined.
2. In line with the previous point, if by hydrological application the authors mean "hydrological models", thus: Which models are they? at what scales? representing which physical processes?
3. The scientific advancement is not made clear. This is also reflected in the poorly developed Conclusions section. What new facets does this work offer? What are the hydrological novelties? Where does this leave us?
4. There is an aporia (a logical contradiction) in the methodology that the authors themselves present. If, according to the manuscript under review, the WRF-SAAG model was not designed to simulate singular events but rather hydroclimatic features in South America (lines 111, 214, and 402), why does the article evaluate the performance of the simulations against daily events of precipitation, maximum temperature, and minimum temperature?
5. Following the same line of reasoning, according to Dominguez et al. (2024), the WRF-SAAG runs use a reanalysis product (ERA5) as their initial and boundary conditions, which by definition represents the best snapshot of weather conditions at a specific place and time (Kalnay et al., 1996). Furthermore, Dominguez et al. (2024 - see Fig. 4) present an evaluation of singular events (peak precipitation hour), comparing simulations (Nov. 2018 to Mar. 2019) with station measurements, GPM-IMERG, and ERA5.

**SPECIFIC COMMENTS**

**Title**
It should reflect which hydrological application the authors aim to address, ideally indicating the process and its temporal scale. I recommend incorporating the study area (i.e., continental Chile).

**Abstract**
L6. State explicitly that you will evaluate daily maximum and minimum temperature. This should be made clear in the abstract.

L18. If you are going to use a hydrological model, please declare this in the sentence where you reference the methodology used.

**Introduction**
L37. You use the expression "Satellite-based products," which, for example, is not the case for CR2MET or RF-MET. Change to "Gridded products."

L62/63. The sentence should conclude with at least one citation.

L72. The authors state that there is "little" information on the performance of the CR2MET and RF-MET products; however, in L78 (in the same paragraph), they provide some numbers and cite an article that has already evaluated their performance. There is a logical contradiction in the writing of this paragraph.

**Study domain**
Fig. 1. Where does the precipitation for the catchments used to calculate the runoff coefficient come from? A gridded product? Which one?

L140. Same comment.

**Methods**

**General Comments**
If both CR2MET and RF-MET were constructed using station measurements, does it make sense to compare their performance at those same locations? I agree that the reported errors can be used as a reference for the performance of WRF-SAAG, but many lines of text are wasted on the analysis of these two products. It would be more fruitful to calculate the difference between the grids (WRF-SAAG vs. CR2MET and RF-MET) to visualize substantial differences. Furthermore, the introduction emphasizes the lack of measurements in high-mountain areas, which further highlights the

importance of performing this grid-to-grid comparison; otherwise, the potential of WRF-SAAG remains very limited.

The hydrological application referred to by the authors in the title is hydrological modeling. Even so, this is very general and therefore weak. Models are subject to multiple sources of uncertainty, and parameter calibration can, in turn, yield correct results for the wrong reasons, especially when the only facet being evaluated is the catchment streamflow (Beven 2006; Kirchner 2006).

If the authors decide to incorporate the broad area of hydrological models as their application in a revised version, they should define the working scales, processes, and model types from the beginning (this must be reflected in the methodology). Using a numerical model for the sole purpose of running it does not reveal new advancements in hydrology. For example, does it make sense to apply a temperature-index model, like the TUWmodel, in the Near-North and Far-North macrozones where sublimation can account for more than 70% of the seasonal snowpack (e.g., Ayala et al. 2023)?

*Regarding the Subsections*

Section 4.1 should be entitled "Evaluation of daily precipitation and maximum and minimum temperatures."

In the introduction, the authors emphasize that in mountain areas (e.g., Chile), most stations are located at low altitudes and are scarce, which leverages the use of high-resolution dynamic models (e.g., WRF-SAAG) to capture total precipitation patterns along mountain ranges like the Andes. However, in the proposed methodology, they evaluate the performance of WRF-SAAG using station measurements, the majority of which are located below 3000 m a.s.l. and with a low-density network in the Cordillera.

Perhaps it would be more interesting, given that CR2MET and RF-MET are built with station measurements, to conduct an analysis of the differences (quarterly?) between grids so that the reader can visualize latitudinal and altitudinal discrepancies. In which quarter are the differences smaller (larger)? Why?

In Figure 1.d, the authors show the reference runoff coefficient for each catchment. After clarifying the source of the precipitation, what values does this coefficient yield when using precipitation simulated by WRF-SAAG? CR2MET? RF-MET? How do the time series and the climatological value for each catchment compare? Given the hydroclimatic regime of each catchment, what values would be logical to expect? Are the absolute values of total precipitation reasonable?

After the general comments, if you still wish to incorporate the hydrological model, the following lines should be taken into account,

+ In section 4.2, the hydrological model is poorly presented. TUWmodel is one of the many versions of the original HBV. First, present and cite HBV, then TUWmodel.

+ What daily temperature value do you use as input data for the model? Minimum and maximum? Daily mean? Up to this section, you have stated that you are evaluating the daily maximum and minimum temperatures.

+ Figure 6.c. Except for the river in the Southern macrozone, the model is incapable of simulating the observed annual cycle of the rivers. Is this because WRF-SAAG does not capture the seasonality of precipitation? Could it be that the model does not adequately simulate the dominant physical processes?

***Final suggestion***

After these comments are addressed, I look forward to revisiting the Methods, Results, Discussion, and a richer, more substantive Conclusions section.

**REFERENCES**

Ayala, Á., Schauwecker, S., MacDonell, S., 2023. Spatial distribution and controls of snowmelt runoff in a sublimation-dominated environment in the semiarid Andes of Chile. Hydrology and Earth System Sciences 27, 3463–3484. https://doi.org/10.5194/hess-27-3463-2023

Beven, K., 2006. A manifesto for the equifinality thesis. Journal of Hydrology, The model parameter estimation experiment 320, 18–36. https://doi.org/10.1016/j.jhydrol.2005.07.007

Dominguez, F., Rasmussen, R., Liu, C., Ikeda, K., Prein, A., Varble, A., Arias, P.A., Bacmeister, J., Bettolli, M.L., Callaghan, P., Carvalho, L.M.V., Castro, C.L., Chen, F., Chug, D., Chun, K.P. (Sun), Dai, A., Danaila, L., Rocha, R.P. da, Nascimento, E. de L., Dougherty, E., Dudhia, J., Eidhammer, T., Feng, Z., Fita, L., Fu, R., Giles, J., Gilmour, H., Halladay, K., Huang, Y., Wong, A.M.I., Lagos-Zúñiga, M.Á., Jones, C., Llamocca, J., Llopart, M., Martinez, J.A., Martinez, J.C., Minder, J.R., Morrison, M., Moon, Z.L., Mu, Y., Neale, R.B., Ocasio, K.M.N., Pal, S., Potter, E., Poveda, G., Puhales, F., Rasmussen, K.L., Rehbein, A., Rios-Berrios, R., Risanto, C.B., Rosales, A., Scaff, L., Seimon, A., Somos-Valenzuela, M., Tian, Y., Oevelen, P.V., Veloso-Aguila, D., Xue, L., Schneider, T., 2024. Advancing South American Water and Climate Science through Multidecadal Convection-Permitting Modeling. https://doi.org/10.1175/BAMS-D-22-0226.1

Kalnay, E., Kanamitsu, M., Kistler, R., Collins, W., Deaven, D., Gandin, L., Iredell, M., Saha, S., White, G., Woollen, J., Zhu, Y., Chelliah, M., Ebisuzaki, W., Higgins, W., Janowiak, J., Mo, K.C., Ropelewski, C., Wang, J., Leetmaa, A., Reynolds, R., Jenne, R., Joseph, D., 1996. The NCEP/NCAR 40-Year Reanalysis Project.

Kirchner, J.W., 2006. Getting the right answers for the right reasons: Linking measurements, analyses, and models to advance the science of hydrology. Water Resources Research 42. https://doi.org/10.1029/2005WR004362

---

## Referee Comment (RC3)

**Manuscript Title:** Benchmarking convection-permitting climate simulations for hydrological applications: A comparative study of WRF-SAAG and observation-based products

**General Comments**

This study presents a comprehensive evaluation of the high-resolution, long-term WRF-SAAG climate simulation (2000-2021) against station observations and two gridded meteorological products (CR2MET and RF-MEP) over Chile. The subsequent use of WRF-SAAG outputs to drive a hydrological model (TUW) successfully demonstrates the dataset's utility for hydrological applications. The paper highlights the good performance of WRF-SAAG in capturing precipitation and temperature, particularly in complex mountainous terrain where observational records are sparse. This is a valuable contribution to the regional climate modeling and hydrology communities.

The manuscript is well-structured and the analysis is thorough. However, I have several suggestions for improvement that I believe will enhance the clarity, presentation, and overall impact of the paper. My main suggestions focus on making the writing more concise, improving the presentation of results and data, and expanding the discussion to better guide potential users of these datasets.

**Specific Comments**

**Abstract**

- The abstract, and the paper in general, could be more concise. Please review for opportunities to shorten sentences and state the main findings more directly.
- The sentence on L9-21 is very long and difficult to parse. Please break this down into two or more sentences for clarity.
- In that same sentence, it is unclear which "precipitation products" are being referred to. Please be specific.

**Introduction**

- **L62:** The sentence beginning "As a result, high-resolution atmospheric models…" feels out of place. The preceding text introduces various observational and reanalysis datasets, but there has been no proper introduction to the concept of using high-resolution models as a data source. I suggest moving this sentence to a more logical position, perhaps after L91, where the rationale for using such models is better established.
- **L75:** The text discusses a "high disagreement among CR2MET, RF-MEP, and ERA5," but the RF-MEP dataset has not been properly introduced at this point. Please ensure all datasets are introduced before they are compared or discussed.

- **L105:** This paragraph should more clearly and explicitly state the aims of the study. Currently, it seems the primary goal is to assess WRF-SAAG, but much of the paper also focuses on the inter-comparison of the three gridded products. Clarifying the primary and secondary objectives here would help frame the paper for the reader.

**2. Study Domain**

- **Figure 1b-c:** The colormap used for temperature could be improved. The minimum temperature values around 10°C are close to white, making them difficult to distinguish. Please consider using a different colormap that provides better contrast across the full range of values.

**3. Hydrometeorological Datasets**

- This section introduces four different datasets. To improve clarity and provide an easy reference, I strongly recommend summarizing their key attributes (e.g., spatial resolution, temporal coverage, variables, post-processing methods) in a table.

**5. Results**

- **Figure 2a:** The y-axis scale (currently showing 0.0-1.0) makes the results difficult to read, as all the data points are clustered at the very top of the plots. Please adjust the y-axis scale to a more appropriate range (e.g., 0.5-1.0) to better visualize the differences.
- **Figure 3:** The caption appears to be missing the "WRF-SAAG" label.
- **Evaluation of Gridded Products:** Since CR2MET and RF-MEP both incorporate ground station data using different statistical methods (regression vs. random forest), a brief discussion on the potential sources of uncertainty and discrepancies between these two products would be valuable. Is the disagreement due to the selection of different stations, or the uncertainties inherent in the respective post-processing procedures?
- **Beyond Abstract Metrics:** Figures 2 and 3 provide a good statistical summary, but the information is quite abstract. To give readers a more intuitive understanding of model performance, please supplement the KGE and contingency table metrics with an evaluation of the raw precipitation and temperature fields. For example, providing maps or summary statistics of the seasonal or annual mean biases (e.g., wet/dry bias, warm/cold bias) would be extremely helpful.

**Discussion**

- **L452:** This paragraph provides a good summary of limitations and future work. To increase the impact of the paper, please also provide some specific insights and examples of how the WRF-SAAG and the two observational datasets could be used in practical application studies (e.g., water resource management, agricultural planning, climate change impact assessments). This would provide valuable guidance to other researchers and stakeholders in the region.

---

## Author Comment (AC1)

**Replies to reviewer #1**

**"Benchmarking convection-permitting climate simulations for hydrological applications: A comparative study of WRF-SAAG and observation-based products"**

Sofía Segovia, Pablo A. Mendoza, Miguel Lagos-Zúñiga, Lucía Scaff, and Andreas Prein

We thank the reviewer for his/her time, revision and suggestions to our paper. We provide responses to each individual point below, and how we will address the main comments of the reviewer. For clarity, comments are given in black italics, and our responses are given in plain blue text.

**General Comment:**

The manuscript offers an evaluation of daily total precipitation, maximum daily temperature, and minimum daily temperature variables simulated by the WRF-SAAG convective-permitting numerical model for hydrological applications. It compares these outputs with meteorological station measurements and simulations from the CR2MET and RF-MET products in the conterminous Chile. Furthermore, these variables are used as input data in a conceptual hydrological model (HBV-like) in order to assess the performance of the simulated daily streamflow against observed series in pristine catchments with low glacier cover.

The work addresses a fundamental topic in mountain regions, which is the use of numerical models to supplement the lack of meteorological measurements, especially for precipitation. However, the article has certain structural deficiencies that require manuscript **major revisions**:

1. The title and the body of the text do not specify which hydrological applications are being adressed to (e.g., flood risk, rain-on-snow events, water supply projections, among others - see a non-exhaustive list in Table 1 of Dominguez et al., 2024). The associated temporal scale and hydrological processes are also not defined.

We agree that the original submission lacked the specification of target hydrological applications. In the revised version, we will specify that the proposed framework is applied for simulating diverse hydrological signatures, oriented to mean flow, and extremes, including floods and droughts, through the analysis of hydrological signatures across continental Chile.

Table 1: Hydrologic signatures.

| Signature name      | Signature description                            |
|---------------------|--------------------------------------------------|
| Q/P                 | Runoff coefficient.                              |
| $Q_{mean}$          | Mean daily discharge (mm/d)                      |
| $Q_{JJA}$           | Mean daily discharge of winter days (JJA) (mm/d) |
| $Q_{DJF}$           | Mean daily discharge of summer days (DJF) (mm/d) |
| $oldsymbol{Q}_{95}$ | 95% flow quantile (high flow) (mm/day)           |

| $Q_5$                 | 5% flow quantile (low flow) (mm/d)                         |
|-----------------------|------------------------------------------------------------|
| low_q_freq            | Frequency of low-flow days (< 0.2 times the mean daily     |
|                       | flow) (days/year)                                          |
| high_q_freq           | Frequency of high-flow days (> 9 times the median daily    |
|                       | flow) (days/year)                                          |
| low_q_dur             | Average duration of low-flow events (number of             |
| _                     | consecutive days < 0.2 times the mean daily flow) (d)      |
| high_q_dur            | Average duration of high-flow events (number of            |
|                       | consecutive days > 9 times the median daily flow) (days)   |
| $oldsymbol{Q_{50\%}}$ | Day of year when 50% of the flow volume                    |
| baseflow_index        | Baseflow index (ratio of mean daily baseflow to mean daily |
|                       | discharge, hydrograph separation performed using digital   |
|                       | filter) (-)                                                |
| slope_fdc             | Slope of the flow duration curve (between the log-         |
|                       | transformed 33rd and 66th streamflow percentiles)          |
| stream_elas           | Streamflow precipitation elasticity (sensitivity of        |
|                       | streamflow to changes in precipitation at the annual time  |
|                       | scale) (-)                                                 |

Additionally, we have clarified that the evaluation of precipitation, air temperature and simulated discharge is conducted at the daily time scale, which is consistent with the temporal resolution of the streamflow records used, and serve as a basis for the dominant processes analyzed.

Finally, we have modified the title to reflect the specific applications and study domain:

**"Benchmarking convection-permitting climate simulations for hydrometeorological characterizations: A comparative study of WRF-SAAG and observation-based products in Chile"**

2. In line with the previous point, if by hydrological application the authors mean "hydrological models", thus: Which models are they? at what scales? representing which physical processes?

We have modified the text to specify that (i) only one hydrological model (TUW) is used, (ii) WRF-SAAG and hydrological model simulations are evaluated at the daily time scale, and (iii) average flow conditions, high flow and low flow events, as well as other hydrological signatures such as streamflow precipitation elasticity and the baseflow index, are the target hydrological processes.

3. The scientific advancement is not made clear. This is also reflected in the poorly developed Conclusions section. What new facets does this work offer? What are the hydrological novelties? Where does this leave us?

This work is motivated by the recent publication of the hourly precipitation and temperature simulations for South America (Dominguez et al., 2024), which provide new meteorological

input data for application-relevant research across the continent. One of the areas of interest for the WRF-SAAG community is the evaluation of convection-permitting model output for the characterization of extreme hydrometeorological events through process-based hydrological modeling. This study contributes to this objective by (1) presenting an assessment of WRF-SAAG daily outputs along a hydroclimatically diverse Andean subdomain, which shows comparable performance for replicating hydrometeorological extremes when contrasted against two widely used regional observation-based products -CR2MET (Boisier et al., 2018) and RF-MEP (Baez-Villanueva et al., 2020); and (2) mapping differences between WRF-SAAF, CR2MET and RF-MEP across different hydroclimates. Additionally, this paper expands on previous assessments of WRF precipitation outputs using in-situ observations (e.g., Ikeda et al., 2010; Mendoza et al., 2015) and offline hydrological modeling applications (e.g., Mendoza et al., 2016) conducted in other mountainous regions of the world. The results presented in this paper have shed light on the potential of WRF-SAAG for characterizing diverse hydrological signatures, oriented to mean, and extreme flow and future avenues of research. Importantly, the work presented here has laid the foundation for using kilometer scale model data to study future hydrological changes in the Andes and interpret them based on the performance of such models under current climate conditions. This is essential for increasing resilience to climate change in Andean countries.

We will explicitly emphasize these messages in the revised Introduction and Conclusions sections to highlight the scientific contribution and hydrological relevance of the study.

4. There is an aporia (a logical contradiction) in the methodology that the authors themselves present. If, according to the manuscript under review, the WRF-SAAG model was not designed to simulate singular events but rather hydroclimatic features in South America (lines 111, 214, and 402), why does the article evaluate the performance of the simulations against daily events of precipitation, maximum temperature, and minimum temperature?

While the WRF-SAAG simulations were not designed as a reanalysis dataset (i.e., they do not include data assimilation or spectral nudging), they are nonetheless capable of reproducing historic weather patterns, particularly when these events are influenced by large-scale atmospheric forcing. Therefore, evaluating WRF-SAAG at the daily scale allows us to assess its general skill in representing historical weather variability and the spatial and temporal characteristics of key hydroclimatic variables, even though the system was not designed to reproduce every singular event precisely. However, given the spatial scale of the WRF simulations, they explicitly resolve convective precipitation, even when it is not fully measured by the scare observations, especially in high-elevation zones.

To clarify this point, we will revise the Introduction, Methods, and Discussion sections to explicitly state that the daily-scale evaluation is designed to assess the overall performance of WRF-SAAG in representing historical hydroclimatic variability, rather than to reconstruct individual events exactly, where specific schemes may better reproduce mesoscale events, as previously revised by other authors (e.g., Huang et al., 2024; Lagos-Zúñiga et al., 2024).

5. Following the same line of reasoning, according to Dominguez et al. (2024), the WRF-SAAG runs use a reanalysis product (ERA5) as their initial and boundary conditions, which by definition represents the best snapshot of weather conditions at a specific place and time (Kalnay et al., 1996). Furthermore, Dominguez et al. (2024 - see Fig. 4) present an evaluation of singular events (peak precipitation hour), comparing simulations (Nov. 2018 to Mar. 2019) with station measurements, GPM-IMERG, and ERA5.

We acknowledge that WRF-SAAG uses ERA5 reanalysis data as its initial and boundary conditions, which constrains the simulation to realistic large-scale atmospheric states. While Dominguez et al. (2024) demonstrated the model's capacity to reproduce specific weather events for a particular period, our study has a broader objective: to evaluate WRF-SAAG's overall skill in representing daily hydroclimatic variability across South America. Because the model is forced by ERA5, it can reproduce many large-scale weather events, although smaller-scale or convective processes may be displaced or not captured. This distinction will be clarified in the Introduction, Methods, and Discussion sections, where we have noted the influence of ERA5 boundary conditions, and we explicitly discuss that the evaluation focuses on large-scale features and may not capture all small-scale events.

**Specific Comment:**

**Title**

It should reflect which hydrological application the authors aim to address, ideally indicating the process and its temporal scale. I recommend incorporating the study area (i.e., continental Chile).

We have modified the title following the reviewer's recommendation:

"Benchmarking convection-permitting climate simulations for hydrometeorological characterizations: A comparative study of WRF-SAAG and observation-based products in Chile"

**Abstract**

L6. State explicitly that you will evaluate daily maximum and minimum temperature. This should be made clear in the abstract.

We have been explicit in the abstract that the minimum and maximum daily temperatures are evaluated:

"In this paper, we evaluate the quality of WRF-SAAG daily precipitation and daily maximum and minimum temperature simulations"

L18. If you are going to use a hydrological model, please declare this in the sentence where you reference the methodology used.

The sentence referenced in Line 18 has been revised to explicitly state that a hydrological model is used as part of the methodology. The updated sentence has clarified the role of the hydrological model within the study framework, ensuring transparency from the outset.

**Introduction**

L37. You use the expression "Satellite-based products," which, for example, is not the case for CR2MET or RF-MET. Change to "Gridded products."

We appreciate the reviewer's sentiment. However, the sentence in L37 specifically refers to products derived from satellite observations and, therefore, the term "satellite-based products" is appropriate in that context. The introduction follows a logical structure that first discusses different sources of meteorological data, and the cited text (L37) intentionally refers to satellite-derived datasets to highlight their limitations in complex terrain and high-elevation regions, which is a relevant background for motivating the use of alternative products. Note that CR2MET nor RF-MEP are mentioned in that paragraph, since those products are obtained by combining reanalysis output, topographic descriptors and ground observations.

To avoid any ambiguity, we will revise the paragraph to make a clearer distinction between satellite-based products and gridded blended datasets such as CR2MET and RF-MEP in the following sentences.

*L62/63.* The sentence should conclude with at least one citation.

We have added the references Prein et al. (2023) and Lundquist et al. (2019) associated with the lines:

"As a result, high-resolution atmospheric models can perform similarly (e.g., Prein et al., 2023) or even outperform (e.g., Lundquist et al., 2019) gridded observational products in capturing total precipitation over complex terrain. Additionally, these models offer a physically consistent and spatially continuous representation of precipitation, making them a viable alternative for hydrological modeling applications."

L72. The authors state that there is "little" information on the performance of the CR2MET and RF-MET products; however, in L78 (in the same paragraph), they provide some numbers and cite an article that has already evaluated their performance. There is a logical contradiction in the writing of this paragraph.

We appreciate this suggestion, and we have edited the manuscript to avoid this interpretation. With "limited information", we aimed to state that no systematic assessments of CR2MET and RF-MEP daily precipitation have been conducted using ground measurements as the observational reference. We have reworded the text in that section to clarity this point:

"While these observation-based datasets have been widely used for different applications (e.g., Hernandez et al., 2022; Murillo et al., 2022 in the case of CR2MET, and Chen et al., 2022; Al-Saeedi et al., 2024 in the case of RF-MEP), no systematic assessments of CR2MET

and RF-MEP daily precipitation have been conducted using ground measurements as the observational reference".

Further, the study cited in L78 (Baez-Villanueva et al., 2021) **did not evaluate the products against station observations**; instead, it presented a simple comparison of annual precipitation amounts retrieved from different products across macro-regions, with the aim to assess the impact of the choice of forcing dataset on the regionalization of hydrological model parameters.

**Study domain**

Fig. 1. Where does the precipitation for the catchments used to calculate the runoff coefficient come from? A gridded product? Which one?

In the current version of the preprint, the precipitation used to calculate the runoff coefficient was obtained from the gridded dataset CR2MET v2. However, for the revised version of the manuscript, we will recalculate the runoff coefficient using the three available precipitation products to provide a more robust and comparative assessment. In addition, we will include other hydrological signatures to complement the evaluation of hydrological modeling performance.

L140. Same comment.

The precipitation dataset is CR2MET (v2.5). Please see our previous response.

**Methods - General Comments**

If both CR2MET and RF-MET were constructed using station measurements, does it make sense to compare their performance at those same locations? I agree that the reported errors can be used as a reference for the performance of WRF-SAAG, but many lines of text are wasted on the analysis of these two products. It would be more fruitful to calculate the difference between the grids (WRF-SAAG vs. CR2MET and RF-MET) to visualize substantial differences. Furthermore, the introduction emphasizes the lack of measurements in highmountain areas, which further highlights the importance of performing this grid-to-grid comparison; otherwise, the potential of WRF-SAAG remains very limited.

We have decided to keep the assessment of CR2MET and RF-MEP against station observations for two reasons: (1) the results demonstrate that, despite the precipitation products combine reanalysis data and ground observations, they were not designed to match station measurements perfectly; and (2) the reported errors are valuable for the hydrometeorology community since this is the first assessment against in-situ observations at the daily time scale.

Additionally, we appreciate and agree with the reviewer's recommendation regarding a grid to grid comparison. Therefore, we will include figures with differences between products for different temporal aggregations, with the aim to identify areas with scarce or null available observations.

The hydrological application referred to by the authors in the title is hydrological modeling. Even so, this is very general and therefore weak. Models are subject to multiple sources of uncertainty, and parameter calibration can, in turn, **yield correct results for the wrong reasons**, especially when the only facet being evaluated is the catchment streamflow (Beven, 2006; Kirchner, 2006).

If the authors decide to incorporate the broad area of hydrological models as their application in a revised version, they should define the working scales, processes, and model types from the beginning (this must be reflected in the methodology). Using a numerical model for the sole purpose of running it does not reveal new advancements in hydrology. For example, does it make sense to apply a temperature-index model, like the TUWmodel, in the Near-North and Far-North macrozones where sublimation can account for more than 70% of the seasonal snowpack (e.g., Ayala et al., 2023)?

We appreciate the reviewer's feedback, and we agree that the hydrological modeling application originally presented can be substantially strengthened. In the revised manuscript, we will address this point by expanding the hydrological evaluation (i) by comparing a broader set of hydrological signatures to assess model behavior across different temporal and process-based dimensions, and (ii) by including an analysis of additional model outputs, such as evapotranspiration (ET) and soil moisture. Additionally, we will explicitly declare the TUWmodel limitations, especially in arid regions where turbulent fluxes are not represented by conceptual rainfall-runoff models. One of the specific goals of this research is to test the applicability of WRF-SAAG simulations for hydrological applications starting with simple models as it hasn't been tested yet in continental Chile. To highlight the specific goals, we will explicitly state them in the introduction to settle the readers' expectations.

**Methods - Regarding the Subsections**

Section 4.1 should be entitled "Evaluation of daily precipitation and maximum and minimum temperatures."

We thank the reviewer for this suggestion. The title of Section 4.1 has been modified to: "Evaluation of daily precipitation and maximum and minimum temperatures".

In the introduction, the authors emphasize that in mountain areas (e.g., Chile), most stations are located at low altitudes and are scarce, which leverages the use of high-resolution dynamic models (e.g., WRF-SAAG) to capture total precipitation patterns along mountain ranges like the Andes. However, in the proposed methodology, they evaluate the performance of WRF-SAAG using station measurements, the majority of which are located below 3000 m a.s.l. and with a low-density network in the Cordillera.

We acknowledge the limitations associated with the sparse and low-altitude distribution of meteorological stations in mountainous regions such as the Andes. However, the evaluation of WRF-SAAG and the other gridded products was conducted using the maximum number of available quality-controlled stations, which represents the best observational information currently accessible for continental Chile.

To complement the station-based evaluation and provide additional insights into the representation of those fields in ungauged regions, the revised manuscript will incorporate a direct comparison of the spatial fields from the three meteorological products (CR2MET, RF-MEP and WRF-SAAG). This gridded intercomparison highlights the main differences among products across elevation gradients and complex terrain, particularly in the Andes.

Perhaps it would be more interesting, given that CR2MET and RF-MET are built with station measurements, to conduct an analysis of the differences (quarterly?) between grids so that the reader can visualize latitudinal and altitudinal discrepancies. In which quarter are the differences smaller (larger)? Why?

We agree that a comparison between the gridded products would provide valuable insights, particularly because CR2MET and RF-MEP are derived from station observations while WRF-SAAG originates from a dynamical model. As noted in the previous response, the revised manuscript will include a direct intercomparison of the gridded fields from the three meteorological products to highlight latitudinal and altitudinal differences across Chile. Different temporal aggregations have also been evaluated to identify the main differences among the products.

In Figure 1.d, the authors show the reference runoff coefficient for each catchment. After clarifying the source of the precipitation, what values does this coefficient yield when using precipitation simulated by WRF-SAAG? CR2MET? RF-MET? How do the time series and the climatological value for each catchment compare? Given the hydroclimatic regime of each catchment, what values would be logical to expect? Are the absolute values of total precipitation reasonable?

In the revised version of the manuscript, we will calculate the runoff coefficient – together with other hydrological signatures — using data from the three available products: WRF-SAAG, CR2MET, and RF-MET. This has allowed us to evaluate the consistency among datasets and to quantify how the choice of precipitation and temperature product affects the estimation of runoff and other hydrological signatures.

After the general comments, if you still wish to incorporate the hydrological model, the following lines should be taken into account,

• In section 4.2, the hydrological model is poorly presented. TUWmodel is one of the many versions of the original HBV. First, present and cite HBV, then TUWmodel.

The manuscript has been revised to first present the original HBV model, including appropriate citations, before introducing TUWmodel as a specific version.

• What daily temperature value do you use as input data for the model? Minimum and maximum? Daily mean? Up to this section, you have stated that you are evaluating the daily maximum and minimum temperatures.

For the hydrological model, the daily mean temperature, calculated as the average of the daily minimum and maximum temperatures, is used as input. The manuscript has been revised to explicitly state this in Section 4.2 to ensure consistency with the previously described evaluation of daily minimum and maximum temperatures.

• Figure 6.c. Except for the river in the Southern macrozone, the model is incapable of simulating the observed annual cycle of the rivers. Is this because WRF-SAAG does not capture the seasonality of precipitation? Could it be that the model does not adequately simulate the dominant physical processes?

To provide more insights on the mismatch between simulated and observed runoff seasonalities, we will include additional figures of annual cycles of precipitation and temperature in the revised version of the manuscript.

**Final suggestion**

After these comments are addressed, I look forward to revisiting the Methods, Results, Discussion, and a richer, more substantive Conclusions section.

We thank the reviewer for the constructive feedback and valuable suggestions. All points raised will be address in the revised manuscript, including clarifications in the Methods, Results, and Discussion sections, as well as an expanded and more substantive Conclusions section.

**References:**

- Al-Saeedi, B. A., M. Baez-Villanueva, O., & Ribbe, L. (2024). An optimized representation of precipitation in Jordan: Merging gridded precipitation products and ground-based measurements using machine learning and geostatistical approaches. https://doi.org/10.5194/egusphere-egu24-11510
- Ayala, Á., Schauwecker, S., & MacDonell, S. (2023). Spatial distribution and controls of snowmelt runoff in a sublimation-dominated environment in the semiarid Andes of Chile. *Hydrology and Earth System Sciences*, 27(18), 3463–3484. https://doi.org/10.5194/hess-27-3463-2023
- Baez-Villanueva, O. M., Zambrano-Bigiarini, M., Beck, H. E., McNamara, I., Ribbe, L., Nauditt, A., Birkel, C., Verbist, K., Giraldo-Osorio, J. D., & Xuan Thinh, N. (2020). RF-MEP: A novel Random Forest method for merging gridded precipitation products and ground-based measurements. *Remote Sensing of Environment*, 239. https://doi.org/10.1016/j.rse.2019.111606
- Baez-Villanueva, O. M., Zambrano-Bigiarini, M., Mendoza, P. A., McNamara, I., Beck, H. E., Thurner, J., Nauditt, A., Ribbe, L., & Thinh, N. X. (2021). On the selection of precipitation products for the regionalisation of hydrological model parameters. *Hydrology and Earth System Sciences*, 25(11), 5805–5837. https://doi.org/10.5194/hess-25-5805-2021
- Beven, K. (2006). A manifesto for the equifinality thesis. *Journal of Hydrology*, *320*(1), 18–36. https://doi.org/https://doi.org/10.1016/j.jhydrol.2005.07.007
- Boisier, J. P., Alvarez-Garretón, C., Cepeda, J., Osses, A., Vásquez, N., & Rondanelli, R. (2018). CR2MET: A high-resolution precipitation and temperature dataset for

- hydroclimatic research in Chile. In EGU general assembly conference abstracts (p. 19739).
- Chen, C., He, M., Chen, Q., Zhang, J., Li, Z., Wang, Z., & Duan, Z. (2022). Triple collocation-based error estimation and data fusion of global gridded precipitation products over the Yangtze River basin. *Journal of Hydrology*, 605. https://doi.org/10.1016/j.jhydrol.2021.127307
- Dominguez, F., Rasmussen, R., Liu, C., Ikeda, K., Prein, A., Varble, A., Arias, P. A.,
  Bacmeister, J., Bettolli, M. L., Callaghan, P., Carvalho, L. M. V., Castro, C. L., Chen,
  F., Chug, D., Chun, K. P. S., Dai, A., Danaila, L., da Rocha, R. P., de Lima Nascimento,
  E., ... Schneider, T. (2024). Advancing South American Water and Climate Science
  through Multidecadal Convection-Permitting Modeling. *Bulletin of the American Meteorological Society*, 105(1), E32–E44. https://doi.org/10.1175/BAMS-D-22-0226.1
- Hernandez, D., Mendoza, P. A., Boisier, J. P., & Ricchetti, F. (2022). Hydrologic Sensitivities and ENSO Variability Across Hydrological Regimes in Central Chile (28°–41°S). *Water Resources Research*, 58(9). https://doi.org/10.1029/2021WR031860
- Huang, Y., Xue, M., Hu, X.-M., Martin, E., Novoa, H. M., McPherson, R. A., Liu, C., Ikeda, K., Rasmussen, R., Prein, A. F., Perez, A. V., Morales, I. Y., Ticona Jara, J. L., & Flores Luna, A. J. (2024). Characteristics of Precipitation and Mesoscale Convective Systems Over the Peruvian Central Andes in Multi 5-Year Convection-Permitting Simulations. *Journal of Geophysical Research: Atmospheres*, 129(17), e2023JD040394. https://doi.org/https://doi.org/10.1029/2023JD040394
- Ikeda, K., Rasmussen, R., Liu, C., Gochis, D., Yates, D., Chen, F., Tewari, M., Barlage, M., Dudhia, J., Miller, K., Arsenault, K., Grubišić, V., Thompson, G., & Guttman, E. (2010). Simulation of seasonal snowfall over Colorado. *Atmospheric Research*, *97*(4), 462–477. https://doi.org/https://doi.org/10.1016/j.atmosres.2010.04.010
- Kalnay, E., Kanamitsu, M., Kistler, R., Collins, W., Deaven, D., Gandin, L., Iredell, M., Saha, S., White, G., Woollen, J., Zhu, Y., Chelliah, M., Ebisuzaki, W., Higgins, W., Janowiak, J., Mo, K. C., Ropelewski, C., Wang, J., Leetmaa, A., ... Joseph, D. (1996). The NCEP/NCAR 40-Year Reanalysis Project. *Bulletin of the American Meteorological Society*, 77(3), 437–472. https://doi.org/10.1175/1520-0477(1996)077<0437:TNYRP>2.0.CO;2
- Kirchner, J. W. (2006). Getting the right answers for the right reasons: Linking measurements, analyses, and models to advance the science of hydrology. *Water Resources Research*, 42(3). https://doi.org/https://doi.org/10.1029/2005WR004362
- Lagos-Zúñiga, M., Balmaceda-Huarte, R., Regoto, P., Torrez, L., Olmo, M., Lyra, A., Pareja-Quispe, D., & Bettolli, M. L. (2024). Extreme indices of temperature and precipitation in South America: trends and intercomparison of regional climate models. *Climate Dynamics*, 62(6), 4541–4562. https://doi.org/10.1007/s00382-022-06598-2
- Lundquist, J., Abel, M. R., Gutmann, E., & Kapnick, S. (2019). Our Skill in Modeling Mountain Rain and Snow is Bypassing the Skill of Our Observational Networks. *Bulletin of the American Meteorological Society*, 100(12), 2473–2490. https://doi.org/10.1175/BAMS-D-19-0001.1
- Mendoza, P. A., Clark, M. P., Mizukami, N., Gutmann, E. D., Arnold, J. R., Brekke, L. D., & Rajagopalan, B. (2016). How do hydrologic modeling decisions affect the portrayal of climate change impacts? *Hydrological Processes*, 30(7), 1071–1095. https://doi.org/https://doi.org/10.1002/hyp.10684

- Mendoza, P. A., Rajagopalan, B., Clark, M. P., Ikeda, K., & Rasmussen, R. M. (2015). Statistical Postprocessing of High-Resolution Regional Climate Model Output. *Monthly Weather Review*, *143*(5), 1533–1553. https://doi.org/10.1175/MWR-D-14-00159.1
- Murillo, O., Mendoza, P. A., Vásquez, N., Mizukami, N., & Ayala, Á. (2022). Impacts of Subgrid Temperature Distribution Along Elevation Bands in Snowpack Modeling: Insights From a Suite of Andean Catchments. *Water Resources Research*, 58(12). https://doi.org/10.1029/2022WR032113
- Prein, A. F., Ban, N., Ou, T., Tang, J., Sakaguchi, K., Collier, E., Jayanarayanan, S., Li, L., Sobolowski, S., Chen, X., Zhou, X., Lai, H.-W., Sugimoto, S., Zou, L., Hasson, S. ul, Ekstrom, M., Pothapakula, P. K., Ahrens, B., Stuart, R., ... Chen, D. (2023). Towards Ensemble-Based Kilometer-Scale Climate Simulations over the Third Pole Region. *Climate Dynamics*, 60(11), 4055–4081. https://doi.org/10.1007/s00382-022-06543-3

---

## Author Comment (AC2)

**Replies to reviewer #2**

**"Benchmarking convection-permitting climate simulations for hydrological applications: A comparative study of WRF-SAAG and observation-based products"**

Sofía Segovia, Pablo A. Mendoza, Miguel Lagos-Zúñiga, Lucía Scaff, and Andreas Prein

We thank the reviewer for his/her time, revision and suggestions to our paper. We provide responses to each individual point below, and how we will address the main comments of the reviewer. For clarity, comments are given in black italics, and our responses are given in plain blue text.

This manuscript evaluates precipitation and temperature from WRF-SAAG, CR2MET, and RF-MEP against in-situ stations across continental Chile. The topic is important and relevant. Below are my major and minor comments.

**Major Comment:**

1. Since WRF-SAAG is not expected to reproduce specific events, I am unclear about the motivation for this evaluation. Is WRF-SAAG typically used as forcing for hydrological models to obtain hydrological simulations? If so, is this recommended practice? If not, the value of evaluating individual events is limited. In that case, comparing climatological characteristics across datasets might be more meaningful.

The motivation for our evaluation is not to assess event-scale accuracy, but rather to examine the skill of the WRF-SAAG dataset in representing hydroclimatic variability at the daily time scale. This assessment constitutes a necessary step to determine whether WRF-SAAG provides sufficiently realistic precipitation and temperature fields to be considered for subsequent hydrological modeling experiments. In the revised version of the manuscript, we will make this motivation explicit in the Introduction section, clarifying that the purpose is to assess the general suitability of WRF-SAAG as a meteorological dataset and its potential for hydrological modeling.

Additionally, we will highlight that WRF-SAAG simulations are forced with ERA5 reanalysis, so it is expected to reproduce large-scale precipitation events, and explicitly resolve mesoscale processes such as convective precipitation. WRF simulations are often used in studies involving hydrological modeling (e.g., Wagner et al., 2016) and case study analysis (e.g., Naabil et al., 2017), among others. However, the WRF-SAAG simulations have not been exhaustively evaluated in South America, and their skill as meteorological input forcing for hydrological applications (e.g., Xie et al., 2025) remains unexplored. We will explicitly declare these objectives and opportunities in the introduction.

2. As this is an evaluation study, the manuscript would benefit from clearer take-home messages. For example: Which dataset should be preferred under certain conditions or in specific regions? Are there areas where none of the datasets are recommended?

We agree that the manuscript would benefit from clearer take-home message. In the revised version, we will restructure the Conclusions section to provide a more explicit synthesis of the main findings. This includes identifying the strengths and limitations of each dataset under different hydroclimatic and regional conditions.

These clarifications will be supported by the new analyses included in the revision, such as the intercomparison of gridded precipitation datasets and the evaluation of hydrological signatures, which together allow for more robust and regionally relevant conclusions regarding the suitability of each dataset for hydrometeorological characterizations.

3. Figures 3–5: Consider presenting KGE values as boxplots, since the authors mention medians of the KGE frequently. The spatial maps make it difficult to assess the median.

In the revised version of the manuscript, we will provide additional summary metrics, figures and/or tables to help interpreting the results. Additionally, the figures will be provided in high resolution, so readers can zoom-in to specific regions.

4. Line 285: When precipitation events > 1 mm/day are considered, CR2MET performs best overall, but for thresholds > 5, > 10, and > 20 mm/day, RF-MEP performs best. Could this pattern be influenced by the precipitation distribution? For example, are events between 1-5 mm/day dominant when precipitation events > 1 mm/day? If so, it might cause RF-MEP's superior performance at higher intensities to be diluted when all events > 1 mm/day are included

We agree that the apparent change in relative performance between precipitation products across precipitation thresholds may be influenced by the underlying distribution of precipitation intensities.

To address this, in the revised version of the manuscript we will add and evaluate the number of observed precipitation events at the station level and in each gridded product, to complement the threshold-based analysis. This will help clarify how event frequency and intensity distributions influence the comparative performance among datasets.

4. Line 262: Please specify the formula used to estimate PET and list the variables involved.

PET was calculated using the formulation proposed by Oudin et al. (2005), as implemented in the R package airGR (Coron et al., 2023), considering the latitude of the centroid of each elevation band. In the revised manuscript, the exact equation and all input variables have been included to ensure reproducibility.

**Minor Comments:**

6. Abstract: WRF-SAAG data are available until 2021. Why was the evaluation restricted to 2001–2018?

The analysis period (2001–2018) was limited by the availability of the RF-MEP dataset, which covers the period 1983–2018 (Baez-Villanueva et al., 2021). We explicitly justify the choice of analysis period at the end of section 4.1:

"Importantly, all the assessments presented in this study were conducted for the period April/2001-March/2018, which corresponds to the common temporal coverage of the three forcing datasets."

7. Line 120: Consider adding polygons to the maps to delineate the four geographical units, which would help international readers.

In the current version of the figure, the four geographical units (macrozones) are delineated by horizontal red lines, and the name of each macrozone is indicated on the left side of the panel. We have ensured that this is more clearly described in the figure caption to guide readers' interpretation.

8. The first part of the abstract reads like that WRF-SAAG was evaluated against CR2MET and RF-MEP, but in fact all three datasets were compared against in-situ observations. Please revise for clarity.

We have revised the abstract to clarify that all three datasets—WRF-SAAG, CR2MET, and RF-MEP—were evaluated against in-situ station observations:

"In this paper, we evaluate the quality of WRF-SAAG daily precipitation and daily maximum and minimum temperature simulations using observations from meteorological stations over continental Chile for the period 2001–2018. The results are compared with the performance of two widely used gridded meteorological products – CR2MET and RF-MEP – which combine reanalysis data with in-situ measurements."

9. Line 135: What do "daily observations" refer to? Are they discharge (Q) and precipitation (P)? Does this also include temperature?

We refer to daily discharge observations. We have modified the text to clarify this:

"The selected catchments fulfill the following criteria: (i) at least 80% coverage of daily discharge observations during the period April/2001 – March/2018..."

10. Line 210: It seems the reference should be "1b (1c)" instead of "1c (1d)."

Thanks for catching this! We have corrected the references in the revised manuscript.

11. Line 255: The DEM data should be introduced in the Data section.

Although it is used to derive elevation bands in the analysis, it is not a hydrometeorological dataset and therefore does not fit within the "Hydrometeorological datasets" section.

12. Equation 8: Please clarify what Q and 1/Q represent in the KGE' formulation.

Q is daily discharge, and 1/Q is the reciprocal of daily discharge. We have modified the text as follows to explain this:

"where Q is daily discharge, 1/Q is the reciprocal of daily discharge, and KGE' is the modified Kling-Gupta efficiency (KGE', Kling et al., 2012)".

We have also added the following lines of text in section 4.3:

"Finally, KGE'(Q) (KGE'(1/Q)) denotes the KGE obtained from comparing daily time series of simulated and observed Q (1/Q)".

13. Line 220: When referring to precipitation events > 1 mm/day, do these include events exceeding 5, 10, and 20 mm/day, or are they limited to events between 1-5 mm/day? Please clarify.

In our analysis, we are including all events with magnitude larger than the specified threshold, without limiting them to the next lower limit. We have clarified this in the revised version to avoid ambiguity:

"We used metrics formulated from contingency tables to assess the ability of the datasets to replicate historically observed daily precipitation events exceeding thresholds of 1, 5, 10, and 20 mm/d (i.e., events >1, >5, >10, and >20 mm/d)."

14. Line 297: This section appears to discuss only summer results. Why are "all seasons" mentioned here?

The intention was to highlight that, although Figure 3 focuses on summer (DJF), the Southern macrozone consistently shows high  $KGE_T$  median values ( $\geq 0.65$ ) not only in summer but also across the other seasons.

We have modified the text to clarify this:

"Figure 3 shows the spatial distribution of  $KGE_T$  (and its components) for summer (DJF) daily precipitation estimates from WRF-SAAG, CR2MET, and RF-MEP. The highest  $KGE_T$  values in summer are obtained in the Southern macrozone, where median  $KGE_T$  values reach  $\geq 0.65$ , and remain at or above this level throughout all seasons"

**References:**

Baez-Villanueva, O. M., Zambrano-Bigiarini, M., Mendoza, P. A., McNamara, I., Beck, H. E., Thurner, J., Nauditt, A., Ribbe, L., & Thinh, N. X. (2021). On the selection of precipitation products for the regionalisation of hydrological model parameters. *Hydrology and Earth System Sciences*, 25(11), 5805–5837. https://doi.org/10.5194/hess-25-5805-2021

- Coron, L., Delaigue, O., Thirel, G., Dorchies, D., Perrin, C., Michel, C., Andréassian, V., Bourgin, F., Brigode, P., Le Moine, N., Mathevet, T., Mouelhi, S., Oudin, L., Pushpalatha, R., & Valéry, A. (2023). Suite of GR Hydrological Models for Precipitation-Runoff Modelling, R Package version 1.7.4. https://cran.r-project.org/web/packages/airGR/
- Kling, H., Fuchs, M., & Paulin, M. (2012). Runoff conditions in the upper Danube basin under an ensemble of climate change scenarios. *Journal of Hydrology*, 424–425, 264–277. https://doi.org/10.1016/j.jhydrol.2012.01.011
- Naabil, E., Lamptey, B. L., Arnault, J., Olufayo, A., & Kunstmann, H. (2017). Water resources management using the WRF-Hydro modelling system: Case-study of the Tono dam in West Africa. *Journal of Hydrology: Regional Studies*, *12*, 196–209. https://doi.org/https://doi.org/10.1016/j.ejrh.2017.05.010
- Oudin, L., Hervieu, F., Michel, C., Perrin, C., Andréassian, V., Anctil, F., & Loumagne, C. (2005). Which potential evapotranspiration input for a lumped rainfall-runoff model? Part 2 Towards a simple and efficient potential evapotranspiration model for rainfall-runoff modelling. *Journal of Hydrology*, 303(1–4), 290–306. https://doi.org/10.1016/j.jhydrol.2004.08.026
- Wagner, S., Fersch, B., Yuan, F., Yu, Z., & Kunstmann, H. (2016). Fully coupled atmospheric-hydrological modeling at regional and long-term scales: Development, application, and analysis of WRF-HMS. *Water Resources Research*, *52*(4), 3187–3211. https://doi.org/https://doi.org/10.1002/2015WR018185
- Xie, K., Li, L., Chen, H., & Xu, C.-Y. (2025). Assessing the performance of convection-permitting climate model in reproducing basin-scale hydrological extremes: A western Norway case study. *Journal of Hydrology*, 656, 132989. https://doi.org/https://doi.org/10.1016/j.jhydrol.2025.132989

---

## Author Comment (AC3)

**Replies to reviewer #2**

"Benchmarking convection-permitting climate simulations for hydrological applications: A comparative study of WRF-SAAG and observation-based products"

Sofía Segovia, Pablo A. Mendoza, Miguel Lagos-Zúñiga, Lucía Scaff, and Andreas Prein

We thank the reviewer for his/her time, revision and suggestions to our paper. We provide responses to each individual point below, and how we will address the main comments of the reviewer. For clarity, comments are given in black italics, and our responses are given in plain blue text.

**General Comment:**

This study presents a comprehensive evaluation of the high-resolution, long-term WRF-SAAG climate simulation (2000-2021) against station observations and two gridded meteorological products (CR2MET and RF-MEP) over Chile. The subsequent use of WRF-SAAG outputs to drive a hydrological model (TUW) successfully demonstrates the dataset's utility for hydrological applications. The paper highlights the good performance of WRF-SAAG in capturing precipitation and temperature, particularly in complex mountainous terrain where observational records are sparse. This is a valuable contribution to the regional climate modeling and hydrology communities.

The manuscript is well-structured and the analysis is thorough. However, I have several suggestions for improvement that I believe will enhance the clarity, presentation, and overall impact of the paper. My main suggestions focus on making the writing more concise, improving the presentation of results and data, and expanding the discussion to better guide potential users of these datasets.

**Specific Comment:**

**Abstract**

• The abstract, and the paper in general, could be more concise. Please review for opportunities to shorten sentences and state the main findings more directly.

We will revise the abstract and the rest of the manuscript to shorten the text and advocate for conciseness.

• The sentence on L9-21 is very long and difficult to parse. Please break this down into two or more sentences for clarity.

With the modifications proposed for the paper, we are going to restructure the abstract, summarizing the main results and clarifying the writing.

• In that same sentence, it is unclear which "precipitation products" are being referred to. Please be specific.

We referred to the three precipitation datasets. We have modified the text to reflect that change:

"We found that, although the three precipitation datasets (i.e., WRF-SAAG, CR2MET and RF-MEP) ..."

**Introduction**

• L62: The sentence beginning "As a result, high-resolution atmospheric models..." feels out of place. The preceding text introduces various observational and reanalysis datasets, but there has been no proper introduction to the concept of using high-resolution models as a data source. I suggest moving this sentence to a more logical position, perhaps after L91, where the rationale for using such models is better established.

We agree with this reviewer that the alluded text was out of place. Hence, we have moved it to the sixth paragraph of the introduction:

"During the last decade, convection-permitting climate models (CPCMs) have become increasingly popular (Lucas-Picher et al., 2021) because they offer an enhanced representation of precipitation (e.g., Fosser et al., 2020) with the potential to outperform gridded observational products in capturing total precipitation over complex terrain (Lundquist et al., 2019). Additionally, CPCMs do not rely on cumulus parameterizations – detected as an important source of errors in regional climate modeling –, improving land-atmosphere interactions (Prein et al., 2015). CPCMs also offer the opportunity to advance hydrometeorological understanding at kilometer-scale resolution, and have been used for a myriad of purposes, including snowpack analysis (Ikeda et al., 2021), cloud band detection(Zilli et al., 2024), and flood studies (Li et al., 2022) over continental domains (e.g., Liu et al., 2025). In particular, CPCMs offer a physically consistent and spatially continuous representation of precipitation, making them a viable alternative for process-based hydrological modeling applications."

• L75: The text discusses a "high disagreement among CR2MET, RF-MEP, and ERA5," but the RF-MEP dataset has not been properly introduced at this point. Please ensure all datasets are introduced before they are compared or discussed.

In the revised manuscript, we have introduced RF-MEP:

"For example, Boisier et al. (2018) created the gridded meteorological product CR2MET based on the combination of in-situ observations and ERA5 (Hersbach et al., 2020) reanalysis outputs, whereas Baez-Villanueva et al. (2020) developed the Random Forest based MErging Procedure (RF-MEP) for precipitation estimation, which consists of the combination of observational data, meteorological products (e.g., ERA5 reanalysis) and topographic covariates."

• L105: This paragraph should more clearly and explicitly state the aims of the study. Currently, it seems the primary goal is to assess WRF-SAAG, but much of the paper also focuses on the inter-comparison of the three gridded products. Clarifying the primary and secondary objectives here would help frame the paper for the reader.

The paragraph has been revised to clearly and explicitly state the study objectives, distinguishing the primary aim. We have explicitly stated the general scientific questions and the secondary objectives in the revised version of the manuscript.

"The general objective of this study is to evaluate the ability of the WRF-SAAG simulations to represent daily precipitation and maximum and minimum temperatures over continental Chile, and to assess their potential as a meteorological forcing dataset for simulating hydrological signatures, by comparing them with observation-based meteorological products. To achieve this, four specific objectives are addressed: (i) to evaluate the performance of daily meteorological series from WRF-SAAG and observation-based meteorological products for precipitation, maximum temperature, and minimum temperature, against meteorological station observations; (ii) to assess the ability of the precipitation datasets to replicate daily precipitation events of different magnitudes at meteorological stations; (iii) to identify the regions and climatic conditions where the main differences among the three meteorological datasets are observed; and (iv) to analyze the ability of the meteorological datasets to reproduce hydrological signatures associated with mean flow and extreme runoff events in catchments across Chile, using a conceptual hydrological model."

**Study Domain**

• Figure 1b-c: The colormap used for temperature could be improved. The minimum temperature values around 10°C are close to white, making them difficult to distinguish. Please consider using a different colormap that provides better contrast across the full range of values.

In the revised version of the manuscript, we will update the colormap to one that provides better contrast across the full range of temperature values, ensuring that both low and high temperatures are clearly distinguishable in Figures 1b–c.

**Hydrometeorological Datasets**

• This section introduces four different datasets. To improve clarity and provide an easy reference, I strongly recommend summarizing their key attributes (e.g., spatial resolution, temporal coverage, variables, post-processing methods) in a table.

In the revised version of the manuscript, we will incorporate a summary table with meteorological data sets:

Table 1: Meteorological datasets.

| Data set      | Spatial resolution | temporal resolution | Period        | Variable                                       | Reference                                 |
|---------------|--------------------|---------------------|---------------|------------------------------------------------|-------------------------------------------|
| WRF-SAAG      | 4 km               | hourly              | 2000-2021     | Precipitation;
Temperature                  | Dominguez et al. (2024)                   |
| CR2MET (v2.5) | 0.5° × 0.5         | daily               | 1960-2021     | Precipitation; Minimum and maximum temperature | Boisier et al. (2018)                     |
| RF-MEP (v2)   | 0.5° × 0.5         | daily               | 1983–
2018 | Precipitation                                  | Baez-Villanueva
et al. (2020,
2021) |

**Results**

• Figure 2a: The y-axis scale (currently showing 0.0-1.0) makes the results difficult to read, as all the data points are clustered at the very top of the plots. Please adjust the y-axis scale to a more appropriate range (e.g., 0.5-1.0) to better visualize the differences.

We agree that the current y-axis scale (0.0-1.0) compresses the data points at the top of the plots, making it difficult to distinguish differences. In the revised version, we will add a zoomed view of the first column of Figure 2a, adjusting the y-axis scale (0.5-1.0) to better visualize the variations among data points.

• Figure 3: The caption appears to be missing the "WRF-SAAG" label.

In the original submission, the caption included "WRG-SAAG" instead of "WRF-SAAG", and we have corrected the text accordingly. Thanks for catching this!

• Evaluation of Gridded Products: Since CR2MET and RF-MEP both incorporate ground station data using different statistical methods (regression vs. random forest), a brief discussion on the potential sources of uncertainty and discrepancies between these two products would be valuable. Is the disagreement due to the selection of different stations, or the uncertainties inherent in the respective post-processing procedures?

We agree that a discussion of the potential sources of uncertainty and discrepancies between CR2MET and RF-MEP would add value to the manuscript. In the revised version, we will include a brief discussion in the Discussion section addressing how differences in station selection and the statistical methods used for post-processing (regression vs. random forest) can contribute to differences between the two gridded products.

• Beyond Abstract Metrics: Figures 2 and 3 provide a good statistical summary, but the information is quite abstract. To give readers a more intuitive understanding of

model performance, please supplement the KGE and contingency table metrics with an evaluation of the raw precipitation and temperature fields. For example, providing maps or summary statistics of the seasonal or annual mean biases (e.g., wet/dry bias, warm/cold bias) would be extremely helpful.

In the revised version of the manuscript, we will add maps comparing the gridded precipitation and temperature fields at seasonal and annual scales, together with differences among the three datasets. Additionally, we will include a summary of performance metrics—either in tables or new figures, such as a Taylor diagram—to provide a more intuitive assessment of model performance. These additions will be presented in the Supplementary Material to complement the main statistical analyses.

**Discussion**

• L452: This paragraph provides a good summary of limitations and future work. To increase the impact of the paper, please also provide some specific insights and examples of how the WRF-SAAG and the two observational datasets could be used in practical application studies (e.g., water resource management, agricultural planning, climate change impact assessments). This would provide valuable guidance to other researchers and stakeholders in the region.

In response to this comment and with the aim to improve the organization of the ideas, we have divided the Discussion section into subsections, including one entitled "Limitations and future work". We have added the following text within that subsection:

"Future comparative assessments between WRF-SAAG and other gridded products could incorporate the effects of parameter equifinality on hydrological model simulations (e.g., Muñoz-Castro et al., 2023), examine model structural uncertainty (e.g., Saavedra et al., 2022), compare parameter regionalization results for streamflow prediction in ungauged basins (e.g., Baez-Villanueva et al., 2021), and conduct drought propagation analyses (e.g., Lema et al., 2025). Finally, the WRF-SAAG dataset could serve as a baseline for future climate change impact assessments aligned with the efforts of the SAAG community. Recenty, Liu et al. (in preparation) conducted a climate perturbation experiment with the WRF model using the Pseudo Global Warming (PGW) approach (Hara et al., 2008; Kawase et al., 2009; Schär et al., 1996), which applies climate perturbations derived from global climate model projections to adjust the reanalysis-based initial and boundary forcings used in the baseline historical regional climate simulations. Since the method assumes that the storm tracks and frequency entering the domain remain the same in both control and future simulations, these could be used to examine changes in hydrologic processes because of shift in thermodynamic conditions."

**References:**

Baez-Villanueva, O. M., Zambrano-Bigiarini, M., Beck, H. E., McNamara, I., Ribbe, L., Nauditt, A., Birkel, C., Verbist, K., Giraldo-Osorio, J. D., & Xuan Thinh, N. (2020).

- RF-MEP: A novel Random Forest method for merging gridded precipitation products and ground-based measurements. *Remote Sensing of Environment*, 239. https://doi.org/10.1016/j.rse.2019.111606
- Baez-Villanueva, O. M., Zambrano-Bigiarini, M., Mendoza, P. A., McNamara, I., Beck, H. E., Thurner, J., Nauditt, A., Ribbe, L., & Thinh, N. X. (2021). On the selection of precipitation products for the regionalisation of hydrological model parameters. *Hydrology and Earth System Sciences*, 25(11), 5805–5837. https://doi.org/10.5194/hess-25-5805-2021
- Boisier, J. P., Alvarez-Garretón, C., Cepeda, J., Osses, A., Vásquez, N., & Rondanelli, R. (2018). CR2MET: A high-resolution precipitation and temperature dataset for hydroclimatic research in Chile. In *EGU general assembly conference abstracts* (p. 19739).
- Fosser, G., Kendon, E. J., Stephenson, D., & Tucker, S. (2020). Convection-Permitting Models Offer Promise of More Certain Extreme Rainfall Projections. *Geophysical Research Letters*, 47(13), e2020GL088151. https://doi.org/https://doi.org/10.1029/2020GL088151
- Hara, M., Yoshikane, T., Kawase, H., & Kimura, F. (2008). Estimation of the Impact of Global Warming on Snow Depth in Japan by the Pseudo-Global-Warming Method. *Hydrological Research Letters*, 2, 61–64. https://doi.org/10.3178/hrl.2.61
- Hersbach, H., Bell, B., Berrisford, P., Hirahara, S., Horányi, A., Muñoz-Sabater, J., Nicolas, J., Peubey, C., Radu, R., Schepers, D., Simmons, A., Soci, C., Abdalla, S., Abellan, X., Balsamo, G., Bechtold, P., Biavati, G., Bidlot, J., Bonavita, M., ... Thépaut, J. N. (2020). The ERA5 global reanalysis. *Quarterly Journal of the Royal Meteorological Society*, 146(730), 1999–2049. https://doi.org/10.1002/qj.3803
- Ikeda, K., Rasmussen, R., Liu, C., Newman, A., Chen, F., Barlage, M., Gutmann, E., Dudhia, J., Dai, A., Luce, C., & Musselman, K. (2021). Snowfall and snowpack in the Western U.S. as captured by convection permitting climate simulations: current climate and pseudo global warming future climate. *Climate Dynamics*, 57(7–8), 2191–2215. https://doi.org/10.1007/s00382-021-05805-w
- Kawase, H., Yoshikane, T., Hara, M., Kimura, F., Yasunari, T., Ailikun, B., Ueda, H., & Inoue, T. (2009). Intermodel variability of future changes in the Baiu rainband estimated by the pseudo global warming downscaling method. *Journal of Geophysical Research*, 114(D24), D24110. https://doi.org/10.1029/2009JD011803
- Lema, F., Mendoza, P. A., Vásquez, N. A., Mizukami, N., Zambrano-, M., Vargas, X., Zambrano-Bigiarini, M., & Vargas, X. (2025). Technical note: What does the Standardized Streamflow Index actually reflect? Insights and implications for hydrological drought analysis. *Hydrology and Earth System Sciences*, 29(8), 1981–2002. https://doi.org/10.5194/hess-29-1981-2025
- Li, Z., Gao, S., Chen, M., Gourley, J. J., & Hong, Y. (2022). Spatiotemporal Characteristics of US Floods: Current Status and Forecast Under a Future Warmer Climate. *Earth's Future*, 10(10). https://doi.org/10.1029/2022EF002700
- Liu, C., Ikeda, K., Prein, A., Scaff, L., Dominguez, F., Rasmussen, R., Huang, Y., Dudhia, J., Wang, W., Chen, F., Xue, L., Fita, L., Lagos-Zúñiga, M., Lavado-Casimiro, W., Masiokas, M., Puhales, F., & Yepes, L. J. (2025). Convection-permitting climate simulations over South America: Experimentation during different phases of ENSO. *Atmospheric Research*, 316, 107936. https://doi.org/https://doi.org/10.1016/j.atmosres.2025.107936

- Lucas-Picher, P., Argüeso, D., Brisson, E., Tramblay, Y., Berg, P., Lemonsu, A., Kotlarski, S., & Caillaud, C. (2021). Convection-permitting modeling with regional climate models: Latest developments and next steps. *WIREs Climate Change*, *12*(6), e731. https://doi.org/https://doi.org/10.1002/wcc.731
- Lundquist, J., Abel, M. R., Gutmann, E., & Kapnick, S. (2019). Our Skill in Modeling Mountain Rain and Snow is Bypassing the Skill of Our Observational Networks. *Bulletin of the American Meteorological Society*, *100*(12), 2473–2490. https://doi.org/10.1175/BAMS-D-19-0001.1
- Muñoz-Castro, E., Mendoza, P. A., Vásquez, N., & Vargas, X. (2023). Exploring parameter (dis)agreement due to calibration metric selection in conceptual rainfall-runoff models. *Hydrological Sciences Journal*. https://doi.org/10.1080/02626667.2023.2231434
- Prein, A. F., Langhans, W., Fosser, G., Ferrone, A., Ban, N., Goergen, K., Keller, M., Tölle, M., Gutjahr, O., Feser, F., Brisson, E., Kollet, S., Schmidli, J., van Lipzig, N. P. M., & Leung, R. (2015). A review on regional convection-permitting climate modeling: Demonstrations, prospects, and challenges. *Reviews of Geophysics*, *53*(2), 323–361. https://doi.org/https://doi.org/10.1002/2014RG000475
- Saavedra, D., Mendoza, P. A., Addor, N., Llauca, H., & Vargas, X. (2022). A multi-objective approach to select hydrological models and constrain structural uncertainties for climate impact assessments. *Hydrological Processes*, *36*(1). https://doi.org/10.1002/hyp.14446
- Schär, C., Frei, C., Lüthi, D., & Davies, H. C. (1996). Surrogate climate-change scenarios for regional climate models. *Geophysical Research Letters*, 23(6), 669–672. https://doi.org/10.1029/96GL00265
- Zilli, M. T., Lemes, M. R., Hart, N. C. G., Halladay, K., Kahana, R., Fisch, G., Prein, A., Ikeda, K., & Liu, C. (2024). The added value of using convective-permitting regional climate model simulations to represent cloud band events over South America. *Climate Dynamics*, 62(12), 10543–10564. https://doi.org/10.1007/s00382-024-07460-3